# Nucleus accumbens controls wakefulness by a subpopulation of neurons expressing dopamine $D_1$ receptors

Yan-Jia Luo[1,2,3,4], Ya-Dong Li[1], Lu Wang [1,2,3], Su-Rong Yang[1,2,3], Xiang-Shan Yuan[1], Juan Wang[1], Yoan Cherasse [5], Michael Lazarus [5], Jiang-Fan Chen[6], Wei-Min Qu[1,2,3] & Zhi-Li Huang[1,2,3]

Nucleus accumbens (NAc) is involved in behaviors that depend on heightened wakefulness, but its impact on arousal remains unclear. Here, we demonstrate that NAc dopamine $D_1$ receptor ($D_1$R)-expressing neurons are essential for behavioral arousal. Using in vivo fiber photometry in mice, we find arousal-dependent increases in population activity of NAc $D_1$R neurons. Optogenetic activation of NAc $D_1$R neurons induces immediate transitions from non-rapid eye movement sleep to wakefulness, and chemogenetic stimulation prolongs arousal, with decreased food intake. Patch-clamp, tracing, immunohistochemistry, and electron microscopy reveal that NAc $D_1$R neurons project to the midbrain and lateral hypothalamus, and might disinhibit midbrain dopamine neurons and lateral hypothalamus orexin neurons. Photoactivation of terminals in the midbrain and lateral hypothalamus is sufficient to induce wakefulness. Silencing of NAc $D_1$R neurons suppresses arousal, with increased nest-building behaviors. Collectively, our data indicate that NAc $D_1$R neuron circuits are essential for the induction and maintenance of wakefulness.

[1] Department of Pharmacology, State Key Laboratory of Medical Neurobiology, School of Basic Medical Sciences; Institutes of Brain Science and Collaborative Innovation Center for Brain Science, Fudan University, Shanghai 200032, China. [2] Institute for Basic Research on Aging and Medicine, School of Basic Medical Sciences,, Fudan University, Shanghai 200032, China. [3] Shanghai Key Laboratory of Clinical Geriatric Medicine,, Fudan University, Shanghai 200032, China. [4] Shanghai Key Laboratory of Psychotic Disorders, Shanghai Mental Health Center, Shanghai Jiao Tong University School of Medicine, Shanghai Jiao Tong University, Shanghai 201108, China. [5] International Institute for Integrative Sleep Medicine (WPI-IIIS), University of Tsukuba, 1-1-1 Tennodai, Tsukuba, Ibaraki 305-8575, Japan. [6] The Institute of Molecular Medicine, School of Optometry and Ophthalmology and Eye Hospital, Wenzhou Medical University, 270 Xueyuan Road, Wenzhou, Zhejiang 325027, China. These authors contributed equally: Yan-Jia Luo, Ya-Dong Li, Lu Wang. Correspondence and requests for materials should be addressed to W.-M.Q. (email: quweimin@fudan.edu.cn) or to Z.-L.H. (email: huangzl@fudan.edu.cn)

The nucleus accumbens (NAc) is the major component of the ventral striatum and has long been studied as a key structure in mediating a variety of neurobiological behaviors including motivation, reward, feeding, learning, and cognition[1]. These higher brain functions operate based on wakefulness[2,3]. However, whether the NAc is involved in controlling wakefulness remains to be elucidated.

Converging clinical and animal studies have revealed sleep–wake alterations in numerous psychiatric disorders associated with dysfunctions of the NAc, such as anxiety, depression, drug abuse, and addiction[4–8], implying a close relationship between the NAc and sleep–wake regulation. Qiu et al.[9] report that nonspecific lesions of the NAc by ibotenic acid result in an increase in wakefulness. In contrast, Tellez et al.[10] performed in vivo electrophysiological recordings in the NAc of rats and found that a large population of NAc neurons decrease their firing rate during slow-wave sleep. In addition, microinjection of dopamine (DA) $D_1$ or $D_2/D_3$ receptor (R) agonists into the NAc increases locomotion and wakefulness[11]. Moreover, very recently, a study showed that optogenetically activating dopaminergic ventral tegmental area (VTA) terminals in the NAc promotes arousal in freely behaving mice[12]. However, the role of the NAc in mediating sleep or wakefulness is controversial. Therefore, roles of NAc in the regulation of sleep–wake cycle and the underlying neural circuits remain to be elucidated.

The NAc is mostly composed of two subtypes of GABAergic projection neurons: one expresses DA $D_1Rs$, and the other expresses $D_2Rs$[13]. Emerging evidence indicates that $D_1R$ and $D_2R$-expressing neurons play complementary and sometimes opposing roles in higher brain functions involving reward, sensitization, feeding, and depression-like behaviors[14–17]. For example, repeated cocaine exposure potentiates output of $D_1R$ neurons but weakens output of $D_2R$ neurons in the NAc[18]. NAc $D_1R$ neuron activation enhances conditional place preference for cocaine and morphine, whereas $D_2R$ neuron activation prevents it[19,20]. In addition, acute cocaine or morphine administration promotes wakefulness[21], showing predominant c-Fos immunoreactivity in $D_1R$ neurons in the NAc[13,22]. These findings suggest that NAc $D_1R$ neurons, but not $D_2R$ neurons, play important roles in functional modulation of arousal-based behaviors. Recently, Oishi et al.[23] find that direct stimulation of the NAc $A_{2A}R/D_2R$-expressing neurons induces remarkable non-rapid eye movement (NREM) sleep. Therefore, we hypothesize that $D_1R$-expressing neurons in the NAc may be important for generation of wakefulness.

In the present study, fiber photometry was employed to investigate the activity of NAc $D_1R$ neurons during spontaneous sleep–wake cycle. Chemogenetic and optogenetic approaches combined with polysomnographic recordings were used to investigate the necessity of NAc $D_1R$ neurons in arousal. Patch-clamp recordings and immunoelectron microscopy were performed to assess the functional connectivity between NAc $D_1R$ neurons and neurons in the midbrain and lateral hypothalamus (LH). We further examined the feeding and nest-building behaviors during NAc $D_1R$ neuron activation and inhibition. Finally, bidirectional modulation of $D_2R$ neurons was used to identify the possible local circuit between NAc two subpopulations in sleep–wake regulation. Our results provide several lines of evidence regarding NAc $D_1R$ neuron circuit diagram for arousal control.

## Results

**Population activities of NAc $D_1R$ neurons increase during wake.** To investigate real-time activity of NAc $D_1R$ neurons across spontaneous sleep–wake cycles of freely moving mice, we used fiber photometry technology[24] to examine calcium signals of NAc $D_1R$ neurons. Following stereotaxic infusion of the Cre-dependent adeno-associated virus (AAV) encoding the fluorescent calcium indicator GCaMP6f (AAV-EF1α-DIO-GCaMP6f) into the right NAc of 3 $D_1R$-Cre mice, we implanted a fiber optic with its tip into the NAc, and electrodes for simultaneously recording electroencephalogram/electromyogram (EEG/EMG) to judge sleep–wake states (Fig. 1a, b). To test the cell phenotype specificity of the viral vectors, we used immunofluorescence staining for substance-P (SP, a marker for $D_1R$ neurons), and found that majority of GCaMP6f expressing cells were SP positive (Fig. 1c). When GCaMP and EEG/EMG signals were recorded in the home cage of mice, we observed that sleep–wake stages were consistently associated with changes in $D_1R$ neuron population activity (Fig. 1d, f); i.e., during NREM sleep, NAc $D_1R$ neurons displayed a lower GCaMP signal than during either wakefulness or rapid eye movement (REM) sleep (Fig. 1d, e). Notably, NAc $D_1R$ cells began to increase neuronal activities before NREM-to-wake and NREM-to-REM transitions and decrease neuronal activities before wake-to-NREM transitions (Fig. 1d, f). These findings clearly indicate a mechanistic framework for the participation of NAc $D_1R$ neurons in the regulation of sleep and wakefulness.

**Chemogenetic activation of NAc $D_1R$ neurons increases wake.** To examine the effect of NAc $D_1R$ neuron activation on electrocortical and behavioral wakefulness, we expressed a Cre-recombinase-enabled chemogenetic excitatory system under the control of the human synapsin promoter (AAV-hSyn-DIO-hM3Dq-mCherry), via AAV injections, in the bilateral NAc of 8 $D_1R$-Cre mice (NAc-hM3Dq mice; Fig. 2a). CNO did not affect sleep-wake profiles of mice transduced with control virus AAV-hSyn-DIO-mCherry (Supplementary Fig. 2a, b). Whole-cell current-clamp recordings of NAc $D_1R$ neurons expressing hM3Dq showed depolarization with clozapine N-oxide (CNO) treatment ($27.8 \pm 1.2$ mV, $n = 8$ cells), which was sufficient to elicit action potentials (Fig. 2b). In addition, intense nuclear c-Fos immunostaining was observed in hM3Dq-expressing (mCherry +) neurons (Fig. 2c) following systemic CNO administration in vivo, indicating strong activation of NAc $D_1R$ neurons by the hM3Dq ligand.

Next, we implanted the mice with EEG/EMG electrodes for sleep–wake monitoring and investigated the effect of NAc $D_1R$ neurons activation in freely moving mice. Either vehicle or CNO was injected intraperitoneally (i.p.) into NAc-hM3Dq mice at 09:00 (2 h after lights on) when mice usually show a high level of spontaneous sleep, which is characterized by a typical light phase hypnogram with long bouts of NREM sleep marked by high EEG delta power and low EMG activity. CNO significantly increased wakefulness with decreases in NREM and REM sleep during the 2 h following administration, as compared with vehicle (Fig. 2d–f). For the total amount of time spent in wakefulness, NREM sleep, and REM sleep during the 2 h post-injection period, CNO at 3 mg/kg resulted in a 98.4% increase in wakefulness, with 93.5% and 68.7% reductions in NREM and REM sleep, respectively (Fig. 2g). EEG power spectra during wakefulness induced by NAc $D_1R$ neuron activation were similar to spectra displayed during spontaneous wakefulness (Fig. 2h). Similarly, when injected during dark period in which mice show a high level of arousal, CNO significantly increased wakefulness (Supplementary Fig. 1a–d), further demonstrating that activation of NAc $D_1R$ neurons prolonged arousal even in the dark (active) period. Moreover, injection of CNO at 3 mg/kg but not 1 mg/kg and vehicle significantly increased locomotor activity (Supplementary Fig. 3a–d), suggesting activation of NAc $D_1R$ neuron-induced behavioral arousal.

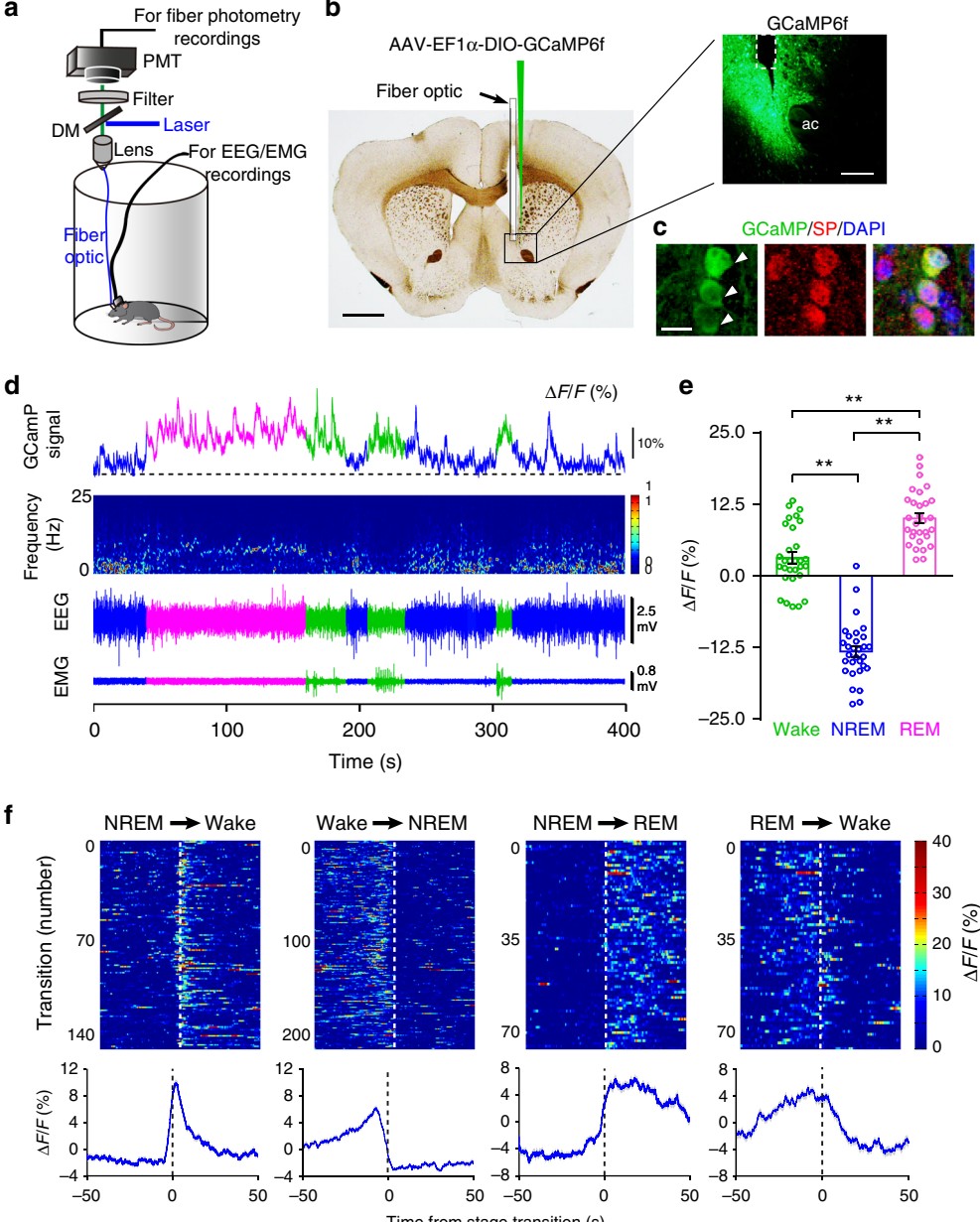

**Fig. 1** Population activities of NAc $D_1R$ neurons across sleep–wake states. **a** Schematic of the fiber photometry setup and in vivo recording configuration. DM dichroic mirror, PMT photomultiplier tube. **b** Unilateral viral targeting of AAV-EF1α-DIO-GCaMP6f into the NAc and the tip of fiber optic above the NAc. Scale bar: 1 mm. Right: Viral expression of GCaMP6f and the placement of fiber-optic probe above the NAc. Scale bar: 200 μm. **c** The 40× confocal images from a NAc $D_1R$ neurons::GCaMP6f brain section immunostained for substance P (SP) and DAPI displaying the colocalization between GCaMP6f-positive neurons (green) and SP (red) cells. White arrowheads highlight NAc $D_1R$ neurons expressing GCaMP6f. Scale bar: 10 μm. **d** Representative fluorescence traces, relative EEG power, and EEG/EMG traces across spontaneous sleep–wake states. $\Delta F/F$ represents change in fluorescence from median of the entire time series. **e** Fluorescence (mean ± s.e.m.) during wake, NREM sleep, and REM sleep for 3 mice, the fluorescence signal is the highest during REM sleep, intermediate during wake, and the lowest during NREM sleep ($n = 3$ mice, 10 sessions per mouse, one-way ANOVA followed by Turkey's post hoc test; $F_{2,87} = 164.9$, $P = 2 \times 10^{-30}$; $P$ (wake–NREM) $= 5 \times 10^{-9}$, (wake–REM) $= 3 \times 10^{-6}$, (NREM–REM) $= 5 \times 10^{-9}$). **f** Fluorescence signals aligned to wake state transitions. Upper panel, individual transitions with color coded fluorescence intensity (NREM–wake, $n = 141$; wake to NREM, $n = 204$; NREM to REM, $n = 78$; REM to wake, $n = 73$). Lower panel, mean (blue trace) ± s.e.m. (gray shading) showing the average calcium transients from all the transitions. **$P < 0.01$

To examine whether NAc $D_1R$ neuron activation would affect sleep-related behavior (nest-building), we removed the old nest from the home cage following administration of vehicle or CNO 3 mg/kg at 9:00, and then provided new paper and quantified nest-building 3 h after injections (Fig. 2i). We found that hM3Dq mice administered with CNO built a poorer nest than control groups did during the light period (Fig. 2j). Moreover, food intake was significantly reduced 2 h after CNO dosing, and this effect lasted for at least 24 h (Fig. 2k), indicating that activation of NAc $D_1R$ neurons decreases food intake.

Taken together, these results indicate that selective activation of NAc $D_1R$-expressing neurons generates wakefulness, along with disrupted nest-building behavior and food intake.

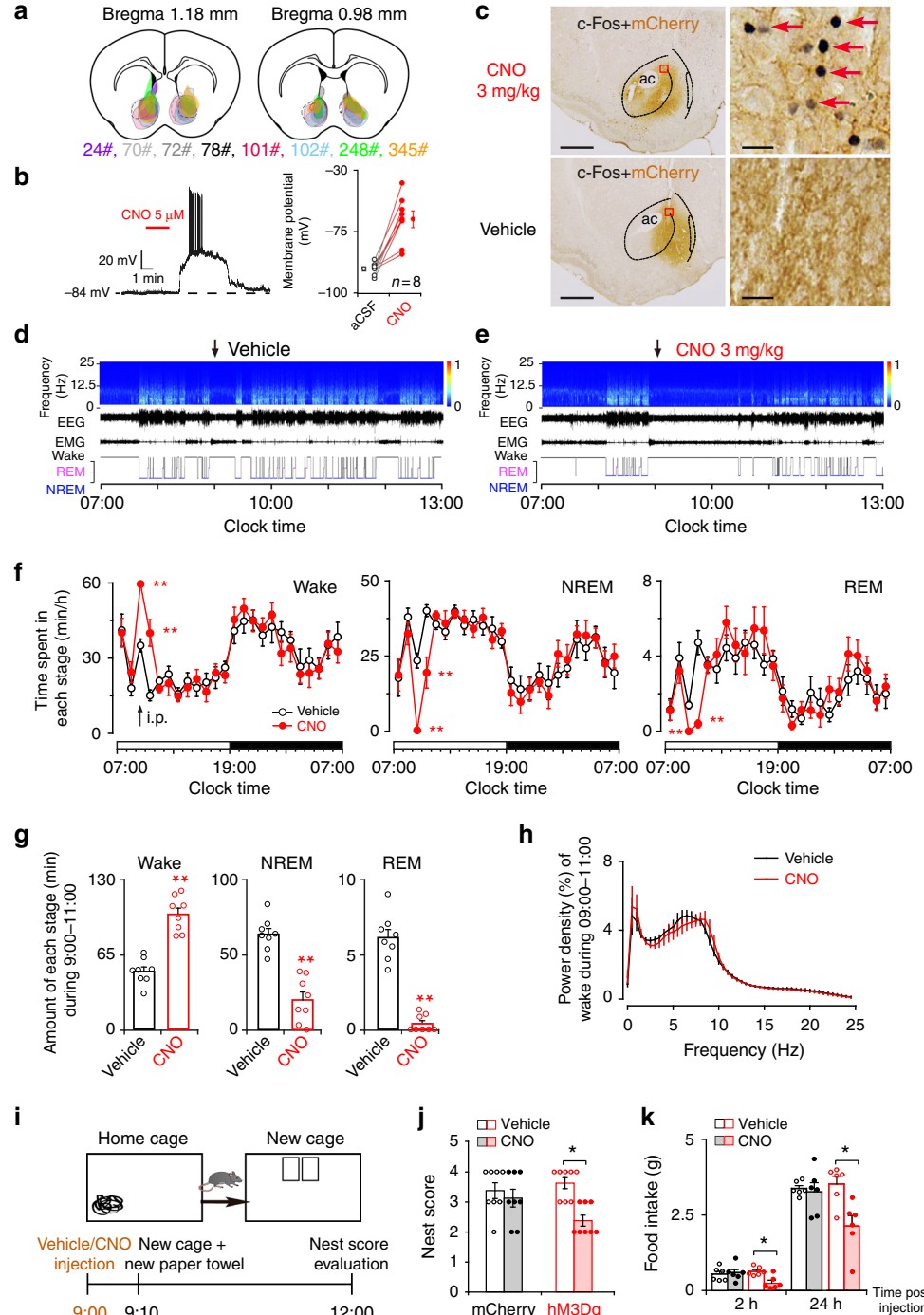

**Fig. 2** Chemogenetic activation of NAc D$_1$R neurons increases wake and suppresses nest-building behavior and food intake. **a** Drawings of superimposed AAV injection sites in the NAc of D$_1$R-Cre mice ($n = 8$, indicated with different colors). **b** Representative voltage traces recorded from an mCherry-expressing neuron during the application of CNO. CNO produced depolarization and firing in an hM3Dq-expressing neuron (left), and induced significant depolarization in hM3Dq-positive D$_1$R neurons (right, $n = 8$ cells from 3 mice, paired $t$ test; $t_7 = 5.8$, $P = 7 \times 10^{-4}$). aCSF artificial cerebrospinal fluid. **c** Representative images of CNO-induced c-Fos (black)/mCherry (brown) colocalization in the NAc. Scale bars: 500 μm. Boxed regions in (**c**) are enlarged in the right panel. Scale bars: 10 μm. **d**, **e** Examples of relative EEG power, EEG/EMG traces, and hypnograms over 6 h following vehicle (**d**) or CNO (**e**) injection at 09:00. **f** Time course changes in wakefulness, NREM sleep, and REM sleep after administration of vehicle or CNO to mice expressing hM3Dq in NAc D$_1$R neurons ($n = 8$, repeated-measures ANOVA; F$_{1,14}$ = 11.9 (wake), 11.6 (NREM), 12.4 (REM); $P = 0.004$ (wake), 0.004 (NREM), 0.003 (REM)). **g** Total time spent in each stage for 2 h after vehicle or CNO injection ($n = 8$, paired $t$ test). **h** EEG power density of wake during the 2 h after vehicle or CNO injection ($n = 8$, paired $t$ test; not statistically significant). **i** Diagram of experiment. **j** Nesting score was assessed in 4 groups ($n = 8$ per group, Wilcoxon matched-pairs signed rank test: mCherry: $Z = 0.6$, $P = 0.5$; hM3Dq: $Z = 2.3$, $P = 0.02$). **k** Systemic application of CNO significantly reduced food consumption ($n = 6$ per group, paired $t$ test). Data represent mean ± s.e.m. *$P < 0.05$, **$P < 0.01$

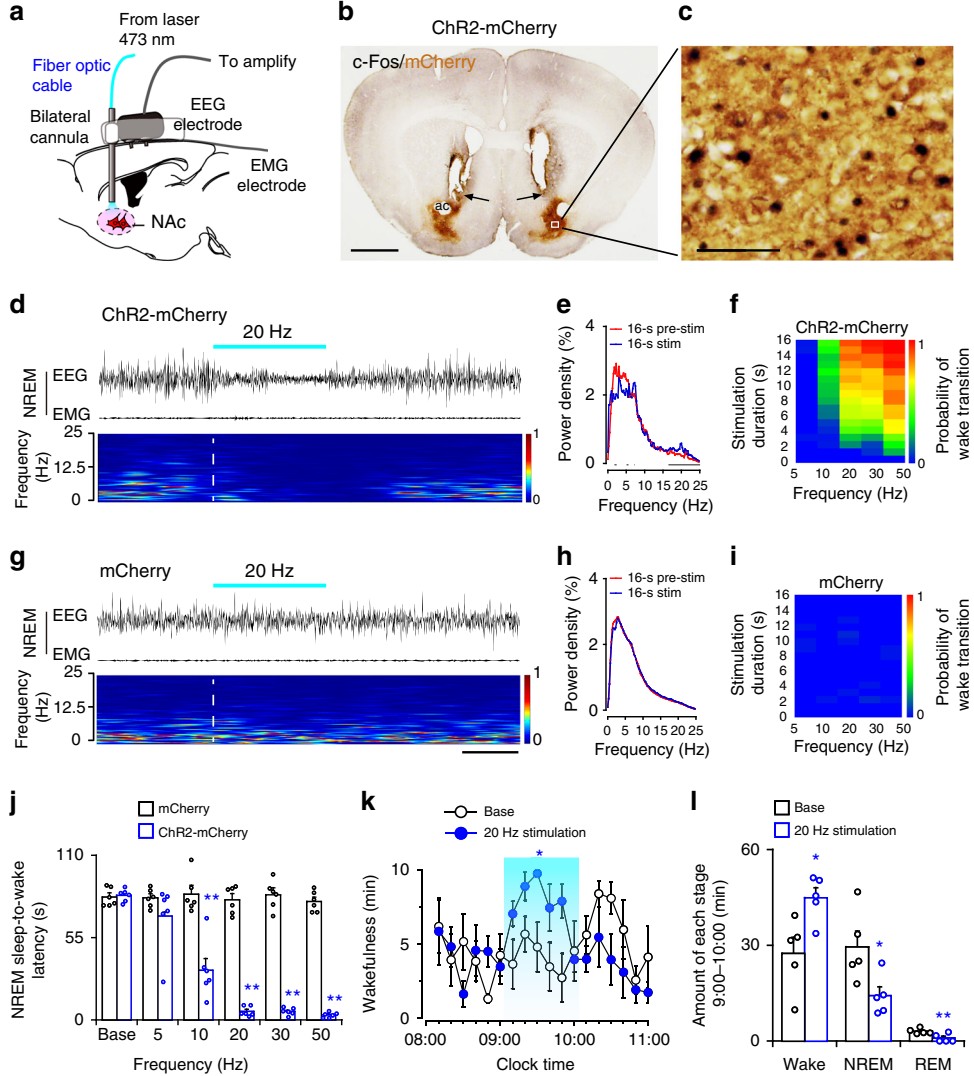

**Fig. 3** Optogenetic activation of NAc $D_1R$ neurons induces a rapid transition from NREM sleep to wakefulness. **a** Diagram of experiment for in vivo optical stimulation. **b** Brain section was stained against mCherry and c-Fos to confirm that the ChR2 protein was expressed in the NAc, and arrows indicate the tip of the optical fiber above the NAc. Scale bar: 1 mm. **c** Higher magnification image of white box in (**b**) indicates abundant c-Fos immunoreactivity in ChR2-mCherry neurons following photostimulation. Scale bar: 50 μm. **d**, **g** Representative EEG/EMG traces, heat map of EEG power spectra show that acute photostimulation (20 Hz/5 ms) applied during NREM sleep induced a transition to wake in a ChR2-mCherry mouse. Scale bar: 8 s. **e**, **h** EEG power (mean) before (red trace) and after (blue trace) onset of photostimulation in ChR2-mCherry (**e**) or mCherry (**h**) mice. Gray lines indicate statistical differences (n = 5 per group; P < 0.05, paired t test). **f**, **i** Heat maps showing the mean probability of NREM sleep-to-wake transitions induced by photostimulation in ChR2-mCherry (**f**) or mCherry (**i**) mice, n = 5 per group. Quantification based on an average of 6–12 stimulations per condition per mouse. Stim stimulation. **j** Latencies of transitions from NREM sleep to wakefulness after photostimulation at different frequencies (n = 6 per group, unpaired t test; Base, $t_{10} = 0.4$, P = 0.7; 5 Hz, $t_{10} = 1.3$, P = 0.2; 10 Hz, $t_{10} = 5.5$, $P = 3 \times 10^{-4}$; 20 Hz, $t_{10} = 18.8$, $P = 4 \times 10^{-9}$; 30 Hz, $t_{10} = 19.5$, $P = 3 \times 10^{-9}$; 50 Hz, $t_{10} = 26.4$, $P = 1 \times 10^{-10}$). **k** Time course of wakefulness during semi-chronic optogenetic experiment (20 Hz/5 ms, 10 s on/20 s off). The blue column indicates the photostimulation period of the stimulation group (n = 5, repeated-measures ANOVA; $F_{1,8} = 10.5$, P = 0.01). **l** Total amounts of each stage in control and photostimulation groups (n = 5, paired t test; $t_4 = 3.7$ (wake), 3.5 (NREM), 5.0 (REM); P = 0.02 (wake), 0.03 (NREM), 0.007 (REM)). Data represent mean ± s.e.m. *P < 0.05, **P < 0.01

## Optogenetic activation of NAc $D_1R$ neurons initiates wake.

Because optogenetics can achieve millisecond timescale control of neuronal activity, we used optogenetic methods to examine the capacity of NAc $D_1R$ neurons to initiate wakefulness. AAV-hSyn-DIO-ChR2-mCherry or AAV-hSyn-DIO-mCherry was delivered to the NAc of $D_1R$-Cre mice (ChR2-mCherry mice or mCherry mice). In vitro experiments confirmed that ChR2 was driven by the blue light with high temporal precision (Supplementary Fig. 4a–g), and optogenetic stimulation evoked inhibitory post-synaptic responses in ChR2-negative neurons (putative $D_2R$/

$A_{2A}R$ neurons; Supplementary Fig. 4h–k), reflecting collateral innervation between NAc neurons.

We applied laser stimulation through optical fibers implanted bilaterally above the NAc 3 weeks after virus injection and quantitatively analyzed EEG/EMG signals recorded simultaneously (Fig. 3a). We stimulated the NAc in vivo in ChR2-mCherry mice with 5 ms pulses of blue light at 20 Hz for 16 s during NREM sleep, and stimulation trials always began 20 s after onset of NREM sleep. After photostimulation, most mCherry-positive neurons were c-Fos-activated in ChR2-mCherry mice

(Fig. 3b, c), indicating activation of ChR2-expressing $D_1R$ neurons by optogenetic stimulation. In addition, 20 Hz stimulation of the bilateral NAc produced immediate transitions from NREM sleep to wakefulness in ChR2-mCherry mice (5.7 ± 1.4 s; Fig. 3d, e, j and Supplementary Movie 1) but not in mCherry mice (80.2 ± 4.2 s; Fig. 3g, h, j and Supplementary Movie 2). In ChR2-mCherry mice, optogenetic stimulation induced immediate cortical activation, which was characterized by a significant decrease in slow-wave activity (a hallmark of sleep pressure)

(Fig. 3d, e). In some cases, the NREM-to-wake transition occurred during the 16 s optical stimulation and was characterized by initial EEG desynchronization without EMG activity followed by head and/or limb movements with increased EMG tone (Supplementary Fig. 5c–f). We analyzed the lag responses and found that in 67.4% of the trials (31 out of 46 trials, 5–12 stimulations per mouse, $n = 6$ mice), the increased EMG occurred at least 1 s after cortical EEG power changed following bilateral optical stimulation at 20 Hz for 16–20 s (Supplementary

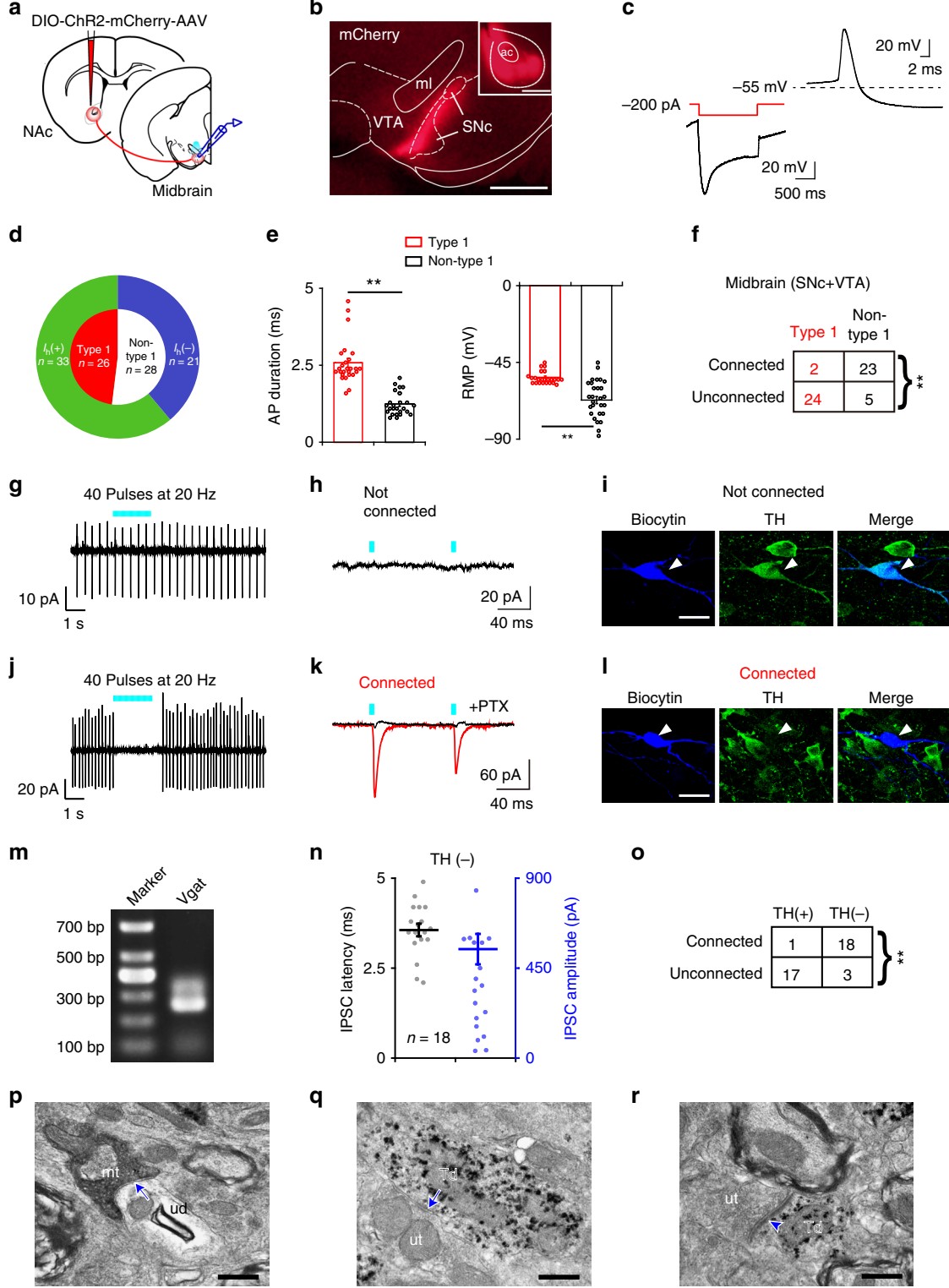

Fig. 5a, b), strongly indicating that NAc $D_1R$ neuron activation-induced arousal was not caused by movement. Moreover, unilateral photostimulation at 20 Hz was also sufficient to induce NREM sleep-to-wake transitions, without obvious unilateral motor effects such as rotation or direction bias (Supplementary Movie 3). Next, by applying 5 ms pulses of photostimulation at 5–50 Hz for 1–16 s during NREM sleep in the light (inactive) period, we found that the probability of transition from NREM sleep to wakefulness increased with frequency and duration of stimulation after the onset of photostimulation in ChR2-mCherry mice, but not in mCherry mice (Fig. 3f, i). These data establish a causal role for NAc $D_1R$ neurons in initiating wakefulness.

**Long-term optical activation of NAc $D_1R$ neurons maintains wake.** To determine whether NAc $D_1R$ neurons maintain wakefulness, we gave photostimulation for 1 h during the light period (09:00–10:00). Sustained activation of NAc $D_1R$ neurons using a semi-chronic optical stimulation procedure (5 ms blue light pulses at 20 Hz for 10 s, every 30 s for 1 h) significantly increased the amount of wakefulness in ChR2-mCherry mice compared with baseline control between 09:00 and 10:00 ($27.5 \pm 5.0$ min at baseline vs. $44.9 \pm 3.1$ min after stimulation, $n = 5$; Fig. 3k, l). Meanwhile, the amount of time spent in both NREM sleep (baseline: $29.5 \pm 5.0$ min; stimulation: $14.2 \pm 2.8$ min) and REM sleep (baseline: $3.1 \pm 0.4$ min; stimulation: $0.9 \pm 0.6$ min) was significantly reduced (Fig. 3l). These findings demonstrate that NAc $D_1R$ neurons are potent in maintaining wakefulness.

**NAc $D_1R$ neurons mainly target midbrain non-DA neurons.** Following AAV-hSyn-DIO-ChR2-mCherry virus injections into the NAc of $D_1R$-Cre mice (Supplementary Fig. 7a, b), we observed the projections to the ventral midbrain. NAc $D_1R$ neuron terminals were distributed mainly in the medial substantia nigra pars compacta (SNc) and VTA and sparsely in the substantia nigra pars reticulata (SNr, Supplementary Fig. 7h–m). To understand better the nature of synaptic connections from NAc $D_1R$ neurons to neurons in different midbrain subregions, we functionally mapped underlying NAc – midbrain connections using electrophysiological recordings in coronal brain slices containing the SNc and VTA of ChR2-mCherry mice (Fig. 4a–o). We classified neurons in the midbrain into two classes based on their electrophysiological properties as described in previous studies[25,26]. We defined type 1 neurons, presumably dopaminergic neurons, when they met the following three criteria: (1) resting membrane potential (RMP) more depolarized than $-70$ mV and fired spontaneous action potentials at frequencies lower than 10 Hz or were quiescent, (2) a pronounced hyperpolarization-activated inward current termed $I_h$ (Fig. 4c, left), and (3) broad action potentials (spike duration >1.3 ms) with a pronounced after-hyperpolarization (Fig. 4c, right).

Neurons that did not satisfy these criteria were identified as non-type 1 neurons. The mean action potential duration for type 1 neurons was significantly longer than that of non-type 1 neurons ($2.6 \pm 0.1$ ms vs. $1.2 \pm 0.1$ ms; Fig. 4e, left), and the RMP of type 1 neurons was more positive ($-53.7 \pm 0.7$ ms vs. $-66.2 \pm 2.1$ mV; Fig. 4e, right). NAc $D_1R$ neurons are believed to release the neurotransmitter GABA upon activation and thus may inhibit postsynaptic targets. Therefore, recorded neurons in the midbrain were considered to have functional connections with NAc $D_1R$ neurons if decreased firing rates and evoked inhibitory post-synaptic currents (IPSCs) (Fig. 4g, h, j, k) were observed when photostimulations activated axonal terminals of NAc $D_1R$ neurons. We found that NAc $D_1R$ neurons disproportionately targeted non-type 1 neurons in the midbrain (Fig. 4f). By adding biocytin to the recording microelectrode solution to allow subsequent immunohistochemical staining for tyrosine hydroxylase (TH), a marker for DA neurons, we confirmed that NAc $D_1R$ neurons mainly targeted TH-negative but not TH-positive neurons in the midbrain (Fig. 4i–l, n, o). These findings indicate that type 1 neurons are TH-positive DA neurons. Notably, since some non-type 1 neurons (7 out of 28 neurons, 25%) were also $I_h$ positive (Fig. 4d), it is not reasonable to distinguish DA neurons and non-DA neurons based on $I_h$ current alone.

Then, as both the SNc and VTA contain DA and non-DA neurons, it is likely that non-DA neurons are primarily GABAergic[25,27]. We detected messenger RNA (mRNA) in three connected neurons using single-cell reverse-transcription PCR (RT-PCR) to confirm vesicular GABA transporter (Vgat) expression (2 out of 3; Fig. 4m). Using immunoelectron microscopy, we found mCherry-immunoreactive terminals establishing symmetric synapses with TH-negative dendrites in the midbrain ($n = 44$ synapses; Fig. 4p), while no synapses were observed between mCherry-immunoreactive terminals and TH-positive dendrites (data not shown). We also found that TH-positive dendrites formed symmetric and asymmetric synapses with mCherry-negative structures ($n = 179$ vs. 43; Fig. 4q, r). These findings indicate that NAc $D_1R$ neurons mostly target GABAergic neurons in the midbrain. Since DA neurons in the midbrain are usually inhibited by neighboring GABAergic neurons[28], we postulated that direct photostimulation of NAc $D_1R$ neurons connected to the midbrain pathway could drive disinhibition of DA neurons.

**NAc $D_1R$ neurons sparsely target NAc-projecting VTA DA neurons.** Neurons in the VTA and SNc send projections to ventral and dorsal parts of the striatum, respectively[27], and we next determined whether there are differences between NAc – SNc and NAc – VTA pathways. By injecting the retrograde tracer CTB-488 into the dorsomedial striatum (DMS) or NAc (Fig. 5a, f) after AAV-ChR2 injection into the NAc, we identified DMS

**Fig. 4** NAc $D_1R$ neurons preferentially send inhibitory inputs to TH-negative neurons in the midbrain. **a** Schematic of experiment. AAV-ChR2 was injected into the NAc of $D_1R$-Cre mice, and the response was recorded in the midbrain (VTA/SNc). **b** Representative image of terminals in the midbrain from NAc $D_1R$ neurons. Scale bars: 500 μm. ac anterior commissure, ml medial lemniscus. **c** Electrophysiological properties of Type 1 neurons in the midbrain. **d** Non-type 1 neurons consist of both $I_h$-positive and $I_h$-negative neurons. **e** Comparison of action potential duration (left) and resting membrane potential (RMP, right) from type 1 and non-type 1 neurons ($n = 26$ type 1 neurons, $n = 28$ non-type 1 neurons, from 17 mice, unpaired $t$ test; action potential duration: $t_{52} = 8.7$, $P = 1 \times 10^{-11}$; RMP: $t_{52} = 5.6$, $P = 9 \times 10^{-7}$). **f** Proportion of connected and unconnected neurons that were identified as type 1 ($n = 26$ type 1 neurons, $n = 28$ non-type 1 neurons, from 17 mice, Chi-square test; $\chi^2 = 30.1$, $P = 2 \times 10^{-8}$). **g–i** Representative images of a non-connected biocytin-filled neuron that was TH-positive. Scale bar: 20 μm. TH tyrosine hydroxylase. **j–l** Typical example of a connected biocytin-labeled neuron that was TH negative and responsive to light stimulation. Scale bar: 20 μm. **m** Representative results from a single-cell RT-PCR reaction confirming the Vgat phenotype of a connected cell. **n** Latency (left axis) and amplitude (right axis) of light-evoked IPSCs in midbrain non-TH neurons. **o** Summary of TH immunocytochemistry for connected and unconnected neurons ($n = 18$ TH (+) neurons, $n = 21$ TH (−) neurons, from 17 mice, Chi-square test; $\chi^2 = 24.9$, $P = 4 \times 10^{-7}$). **p** Electron microscopy image of an mCherry-immunoreactive terminal (mt) that formed a symmetric synapse (arrow) with an unlabeled dendrite (ud). Scale bar: 0.5 μm. **q** A representative image of a TH-positive dendrite (Td) that established a symmetric synapse (arrow) with unlabeled terminals (ut). Scale bar: 0.5 μm. **r** A TH-positive dendrite (Td) that established an asymmetric synapse (arrowhead) with unlabeled terminals (ut). Scale bar: 0.5 μm. **P < 0.01

retrogradely labeled cell bodies in the SNc (Fig. 5b) and NAc retrogradely labeled cell bodies in the VTA that overlapped with fibers containing ChR2 (Fig. 5g). In the SNc, we recorded 12 CTB-488 retrogradely labeled neurons, and none of the labeled neurons was responsive to photostimulation (Fig. 5c, d). These neurons displayed typical electrophysiological properties of DA neurons, and all of them stained positive for TH (Fig. 5e), indicating that the SNc DA neurons were not innervated by NAc $D_1R$ neurons. We then recorded 19 retrogradely labeled VTA neurons,

and all 19 neurons were TH positive (Fig. 5j). Interestingly, only two of these neurons responded to light stimulation with evoked IPSCs (Fig. 5h, i). Taken together, our data show that NAc $D_1R$ neurons do not innervate DA neurons in the SNc, which project to the DMS, and sparsely to DA neurons in the VTA.

**NAc $D_1R$ neurons promote wake via midbrain and LH connections.** To elucidate the neuronal circuits mediating the wake-

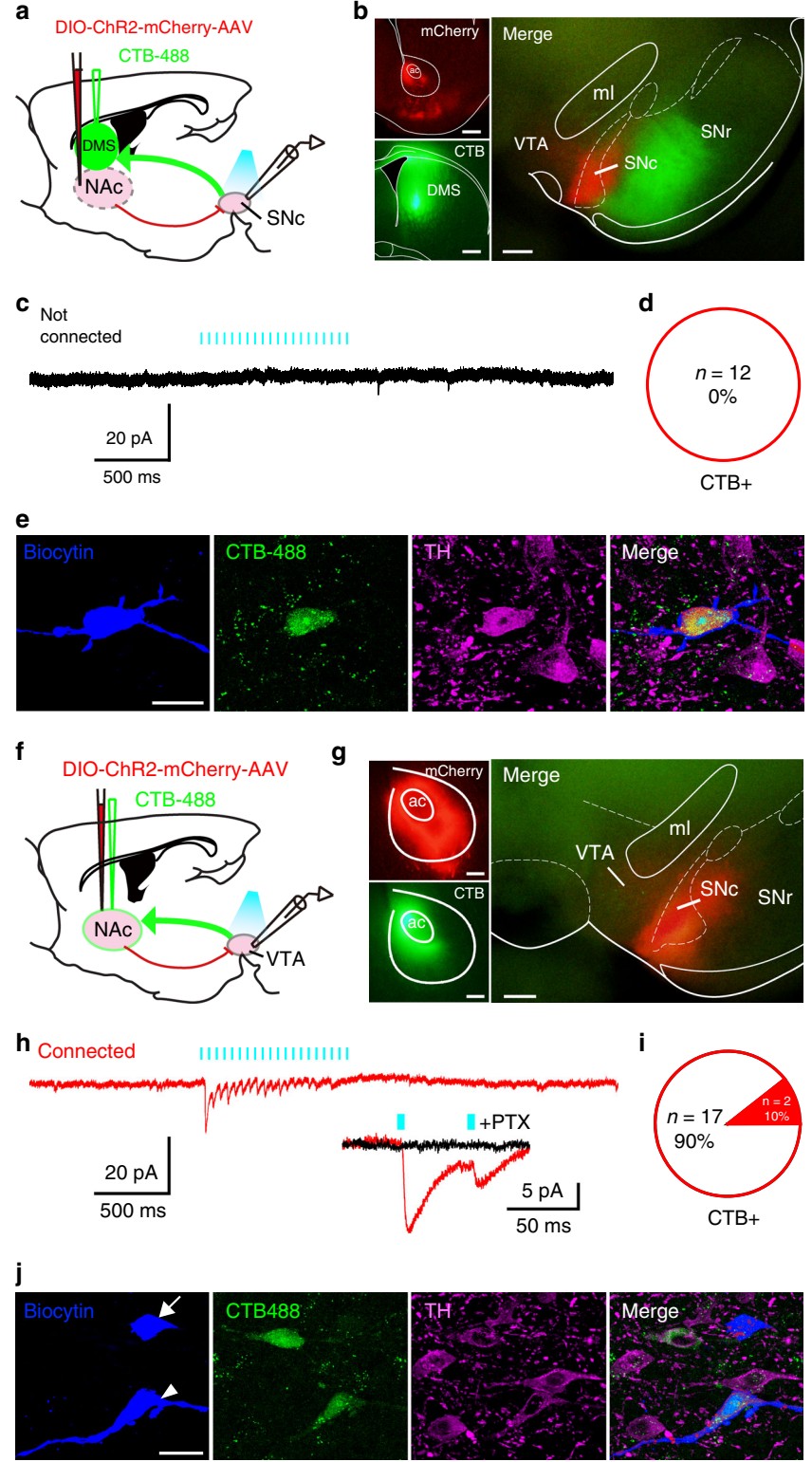

promoting effect of NAc $D_1R$ neurons, ChR2 was expressed in the NAc with optic fibers targeting terminals in the ventral medial midbrain (Fig. 6a, b) or LH (Fig. 6g, h), based on the NAc $D_1R$ neurons projections[14,29] (Supplementary Fig. 7f–m). Brief blue light stimulation at 20 Hz induced an immediate transition from NREM sleep to wakefulness in the midbrain (latency: $5.2 \pm 0.9$ s; Fig. 6c, Supplementary Movie 4) or LH (latency: $6.0 \pm 1.1$ s; Fig. 6i, Supplementary Movie 6). Following yellow light stimulation as a control, no change was observed (Fig. 6d, j and Supplementary Movie 5). Short latencies for sleep-to-wake transition were observed during blue light pulses at the frequencies higher than 20 Hz in the midbrain (Fig. 6e) or LH (Fig. 6k). To identify the effects of long-term stimulation on the wake-promoting effect of NAc $D_1R$ neuron terminals, we applied photostimulation for 1 h during 09:00 to 10:00. Optogenetic stimulation of NAc $D_1R$ neuron terminals in the midbrain (Fig. 6f) or LH (Fig. 6l) increased waking time by 86.4% and 79.4%, respectively, and both with a decrease in NREM sleep and REM sleep compared to controls (Fig. 6f, l). Taken together, these results show that the NAc-midbrain and NAc-LH circuits mediated arousal controlling effect of NAc $D_1R$ neurons (Supplementary Fig. 8a, b).

To test whether NAc $D_1R$ neurons directly inhibit wake-promoting LH GABA neurons[30,31], we crossed $D_1R$-Cre and GAD67-GFP mice to generate a new mouse $D_1R$-Cre::GAD67-GFP. AAV-hSyn-DIO-ChR2-mCherry was injected into the NAc, and in vitro whole-cell recording of LH GABA neurons was used to simultaneously monitor light-evoked IPSCs derived from NAc $D_1R$ neurons afferents (Supplementary Fig. 6a). We found that both green fluorescent protein (GFP)-positive and -negative neurons received inhibitory inputs from NAc $D_1R$ neurons, with no significant difference in connection probability (67% vs. 47%; Supplementary Fig. 6b, d–g). Meanwhile, light-evoked IPSCs latency was less than 5 ms (Supplementary Fig. 6c), indicating the direct connection. Because GAD67 is a marker for GABA, our results suggested that NAc $D_1R$ neurons would target both GABA neurons expressing GAD67 and non-GAD67 neurons including GAD65 GABAergic neurons and non-GABAergic neurons. In hM3Dq study, we found robust c-Fos expression in LH orexin-positive neurons after NAc $D_1R$ neuron activation (Supplementary Fig. 6h, i). Taken together, these findings suggest prominent functional synaptic connectivity between NAc $D_1R$ neurons and LH neurons, and NAc $D_1R$ – LH circuit involving arousal control may be partially through activating LH wake-promoting orexin neurons.

**NAc $D_1R$ neurons are necessary for wakefulness.** To determine whether NAc $D_1R$ neurons are necessary for sustaining wakefulness, we used an inhibitory modified muscarinic G protein-coupled receptor (AAV-hSyn-DIO-hM4Di-mCherry) to selectively silence NAc $D_1R$ neurons (Fig. 7a). In vitro study showed that CNO (5 µM) substantially decreased the number of evoked

action potentials (Fig. 7b). Following i.p. CNO injection at 09:00, total wakefulness during 3 h after CNO administration was significantly decreased, concomitant with an increase in NREM sleep, as compared with vehicle controls ($n = 9$, $P < 0.05$, Fig. 7c, d). In addition, fast Fourier transform analysis of NREM sleep showed an increase in slow delta power in the frequency range of 0–2 Hz (Fig. 7e). Similarly, CNO administration at 21:00 also resulted in a 70.7% increase in NREM sleep during 3 h after injection, which was accompanied by a decrease in wakefulness (Fig. 7f). Moreover, inhibition of $D_1R$ neurons did not change EEG power of NREM and REM sleep (Fig. 7g). Sleep typically occurs in the home environment or nest[12]. To test this, we assessed nest-building behavior. CNO-treated hM4Di mice increased nest building during the test period compared with three control groups (Fig. 7h, i). Finally, inhibition of NAc $D_1R$ neurons produced no noticeable changes in food intake following both 2 h and 24 h periods (Fig. 7j). Taken together, these results indicate that NAc $D_1R$ neurons are essential for the maintenance of physiological arousal.

**NAc $D_2R$ neurons are important to promote NREM sleep.** To test whether arousal-promoting effect of $D_1R$ neurons is partly due to the inhibition of NAc $D_2R$ neurons, we injected hSyn-DIO-hM4Di-mCherry in the bilateral NAc of $D_2R$-Cre mice (Fig. 8a, b) to inhibit $D_2R$ neuron activities. Current-clamp recordings from an hM4Di-expressing $D_2R$ neuron showed decreased action potentials in response to 250 pA current injection during bath application of 5 µM CNO (Fig. 8c). Administration of CNO at 21:00 significantly increased wakefulness by 37.9%, with a decrease in NREM sleep and REM sleep by 97.2% and one-fold during the 2 h post-injection period, respectively, relative to vehicle (Fig. 8d–f). Finally, we examined the sleep-promoting effect after activation of $D_2R$ neurons by CNO in $D_2R$-Cre mice transduced with hM3Dq in the NAc (Fig. 8g, h). In vitro patch-clamp recordings of NAc neurons expressing hM3Dq showed the increased firing response to 180 pA current injection during bath application of 5 µM CNO (Fig. 8i). We found that systemic administration of CNO increased NREM sleep by 2.6-fold and decreased wakefulness by 51.5% during the 2 h post-injection period as compared with vehicle (Fig. 8j–l). These results demonstrate that inhibition of NAc $D_2R$ neurons mimics the effect of activation of accumbal $D_1R$ neurons, and confirm that two subpopulations of $D_1R$ and $D_2R$ neurons in the NAc play opposite roles in sleep–wake control.

**Discussion**

The NAc has long been studied as a key brain region mediating a variety of behaviors[1,32–34] such as locomotion, emotion, cognitive behavior, learning, feeding, action selection, sexual motivation, as well as reward and reinforcement. These behaviors ensure survival and reproduction in nature, which depend on heightened

**Fig. 5** NAc $D_1R$ neurons sparsely target NAc-projecting VTA DA neurons. **a** Schematic of the experimental protocol. Cre-dependent ChR2 was injected into the NAc of $D_1R$-Cre mice, while retrograde tracer CTB-488 was injected into the DMS. Infected terminals (red) were optogenetically activated, and recordings were obtained from retrogradely labeled SNc projection neurons (green). DMS dorsomedial striatum. **b** Representative images of ChR2-mCherry and CTB-488 at the injection sites and their anterogradely and retrogradely labelings in the midbrain slice. Left, scale bars: 500 µm. Right, scale bar: 200 µm. **c** Whole-cell recordings from a DMS projecting SNc neuron did not respond to blue light (5 ms pulses at 20 Hz). **d** No DMS projecting SNc neurons showed connections with NAc $D_1R$ neurons ($n = 12$ cells from 3 mice). **e** Representative confocal images showing a retrogradely labeled SNc neuron that was TH-positive (magenta). Scale bar: 20 µm. **f** Schematic of the experiment. Cre-dependent ChR2 and retrograde tracer CTB-488 were injected into the NAc of $D_1R$-Cre mice, and recordings were obtained from retrogradely labeled VTA projection neurons (green). **g** Images showing ChR2-mCherry and CTB-488 injection in the NAc and anterogradely and retrogradely labeling in the midbrain. Left, scale bars: 200 µm. Right, scale bar: 200 µm. **h** Representative traces of optogenetically generated picrotoxin (PTX, 100 µM)-sensitive postsynaptic currents recorded from a retrogradely labeled VTA neuron. **i** Ten percent of NAc-projecting VTA neurons showed connections with NAc $D_1R$ neurons ($n = 19$ cells from 3 mice). **j** Representative confocal images of VTA slices showing a retrogradely labeled neuron (arrowhead) was TH positive and another patched cell not retrogradely labeled (arrow) was TH negative. Scale bar: 20 µm

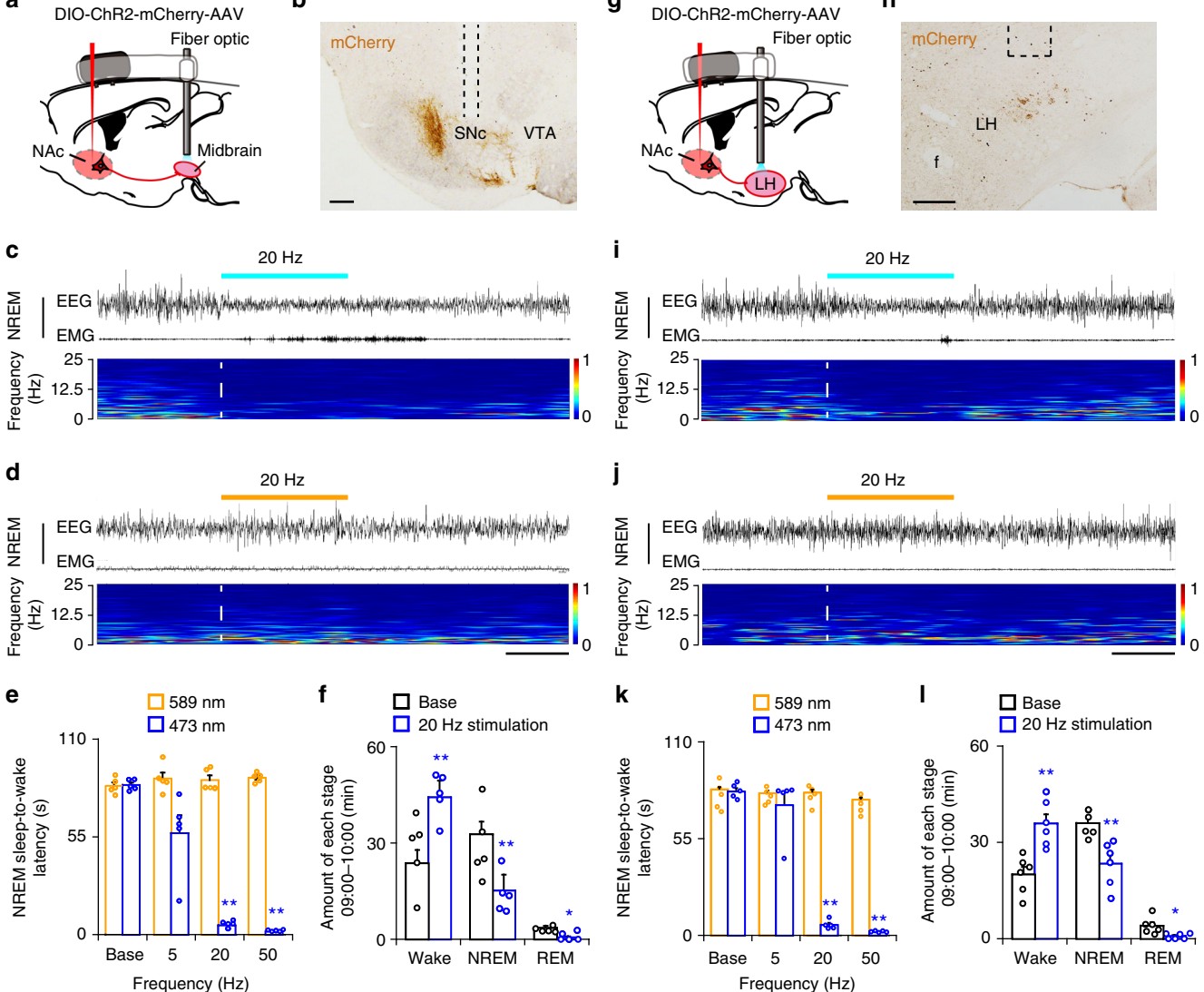

Fig. 6 NAc $D_1R$ neurons control arousal through midbrain and LH pathways. **a**, **g** Schematic diagram showing the location of the optic fiber in the midbrain (**a**) or LH (**g**), and the implanted somnographic electrodes of a $D_1R$-Cre mouse injected with AAV-ChR2-mCherry in the NAc. **b**, **h** Sections stained for mCherry showing ChR2-mCherry-positive terminals in the midbrain (**b**) or LH (**h**), and the cannula trace showing the optic fibers targeting the nuclei. Scale bars: 200 μm; f fornix. **c**, **i** Representative EEG and EMG traces and the corresponding heat map of EEG power spectrum showing that photostimulation in the midbrain (**c**) or LH (**i**) applied during NREM sleep induced a rapid transition to wakefulness. Dashed lines indicate onset of light stimulation. **d**, **j** Representative EEG and EMG recordings and the corresponding heat map of EEG power spectrum showing that yellow light stimulation in the midbrain (**d**) or LH (**j**) failed to affect the sleep–wake pattern. Scale bars: 8 s. **e** Mean latencies of wake transitions during NREM sleep after acute photostimulation at different frequencies ($n = 5$, paired t test; Base, $t_4 = 0.4$, $P = 0.7$; 5 Hz, $t_4 = 2.4$, $P = 0.08$; 20 Hz, $t_4 = 28.5$, $P = 9 \times 10^{-6}$; 50 Hz, $t_4 = 81.8$, $P = 1.3 \times 10^{-7}$). Data analysis was based on an average of 8–12 stimulations per frequency and per mouse. **f** Total amounts of each stage in the control and 20 Hz photostimulation in the midbrain during 09:00–10:00 ($n = 5$, paired t test; $t_4 = 9.5$, $P = 7 \times 10^{-4}$ (wake); $t_4 = 9.4$, $P = 7 \times 10^{-4}$ (NREM); $t_4 = 3.8$, $P = 0.02$ (REM)). **k** Mean latencies of wake transitions during NREM sleep after photostimulation at different frequencies ($n = 5$, paired t test). Data analysis was based on an average of 8–12 stimulations per frequency and per mouse. **l** Total amounts of each stage in the control and 20 Hz photostimulation in the LH during 09:00–10:00 ($n = 6$, paired t test). Data represent mean ± s.e.m. *$P < 0.05$, **$P < 0.01$

arousal. By recording and manipulating the activity of a subpopulation in the NAc with fiber photometry, chemogenetics, optogenetics, behavioral tests, and polysomnographic methods, we demonstrate a prominent contribution of NAc $D_1R$ neurons to the control of wakefulness and the neuron circuits in arousal control.

Compared with traditional methods lacking cellular specificity, the recently developed optogenetic and chemogenetic approaches enable us to manipulate the activity of specific neurons, thus providing powerful tools to examine the roles of different NAc neuron subpopulations in sleep–wake regulation. We here show

that activation of NAc $D_1R$-expressing neurons induces behavioral arousal, while activation of $D_2R$-expressing neurons strongly promotes sleep. Opposite roles of different NAc cell types in regulation of sleep and wakefulness may depend on their projection patterns, in which only $D_1R$ neurons directly project to the midbrain and LH[17,35].

Within the midbrain, there are complex neural circuits involving dopaminergic, GABAergic neurons, and glutamatergic neurons. Anatomic studies have suggested that the NAc projects to all, and provides proportionally more input to DA compared with non-DA neurons[36]; our data clearly showed predominant

functional connections between NAc $D_1R$ neurons and non-dopaminergic neurons, which was consistent with previous optogenetic circuit mapping[25,37]. Targeting of non-DA neurons in the midbrain by NAc-$D_1R$ neurons was further confirmed by our ultrastructural observations of preferential synapses on non-

DA neurons with immunoelectron microscopy. Collectively, inhibitory input from NAc $D_1R$ neurons to the midbrain disinhibits dopamine neurons and promotes wakefulness.

Although dopaminergic cells in the midbrain do not change average firing rates during the sleep–wake cycle[38,39], their

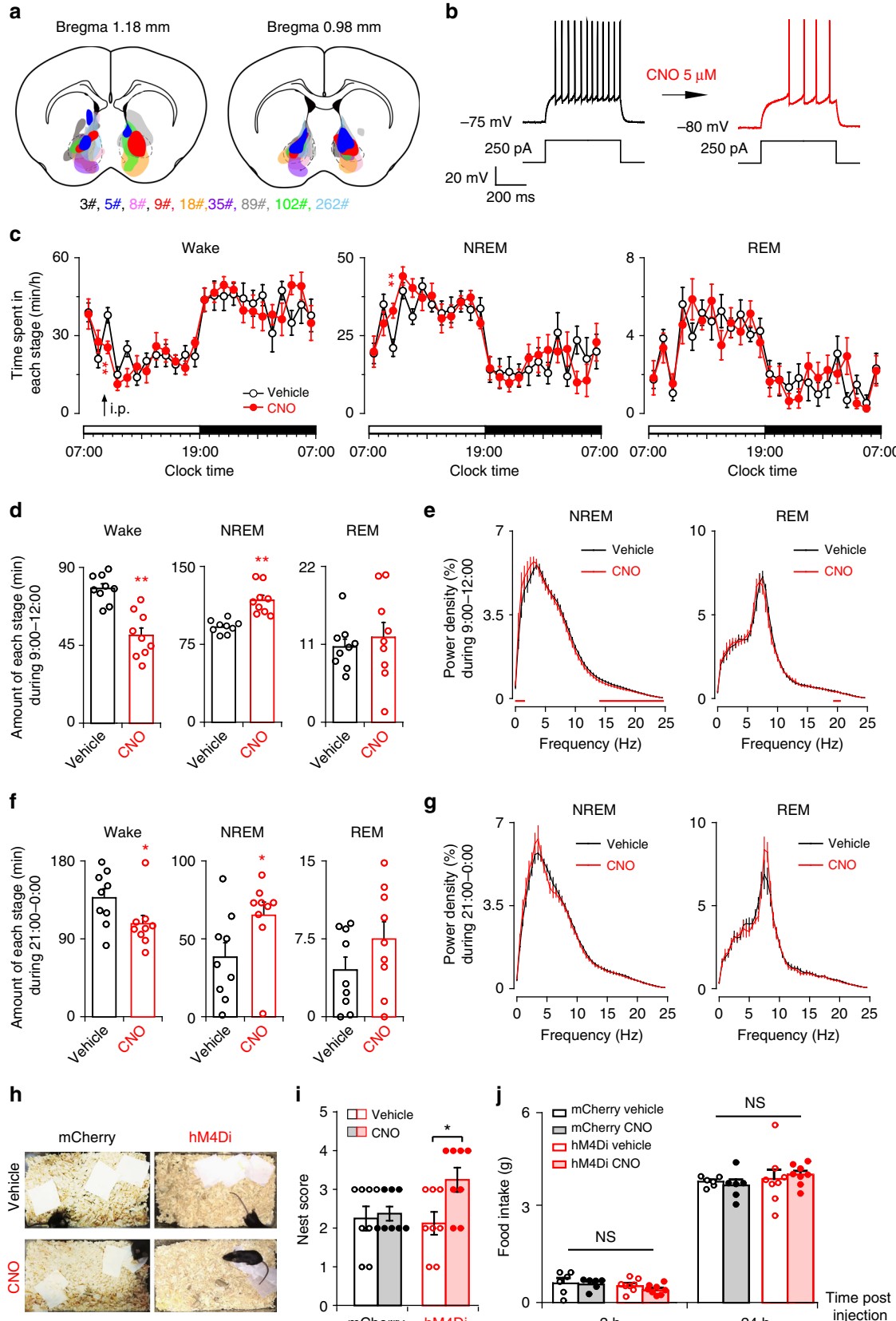

bursting activity is associated with DA release in the NAc and dorsal striatum, respectively[40]. Recently, a study showed that dopaminergic neurons in the VTA are essential for initiation and maintenance of wakefulness, which is mostly mediated by NAc projections[40]. The NAc and VTA are reciprocally connected, forming a NAc/GABA-midbrain/GABA-VTA/DA-NAc/GABA loop that controls wakefulness (Supplementary Fig. 8a, b). Our data support a model wherein activating an inhibitory projection from the NAc to the midbrain promotes arousal through inhibition of GABA neurons, which causes disinhibition of DA neurons to increase DA release in the NAc. This might be a positive feedback process, because the released DA acts on $D_1R$ and $D_2R$ to excite $D_1R$ neurons and inhibit $D_2R$ neurons. However, the endogenous somnogen adenosine increases with prolonged wakefulness and excites $A_{2A}R$ to initiate sleep, which presumably prevents overexcitation[2,3,41].

The LH is a richly heterogeneous structure, containing a number of genetically distinct cell populations, including GABAergic, glutamatergic, and neuropeptide-producing neurons, which are known to have diverse roles in regulating food intake, and sleep–wakefulness[42,43]. Thus, it is critical to identify which LH neurons receive inhibition from NAc $D_1R$ neurons. We found that NAc $D_1R$ neurons innervated both GAD67-positive and GAD67-negative neurons in the LH, showing no distinct preference, and NAc $D_1R$ – LH circuit involving arousal control may be partially through activating LH wake-promoting orexin neurons. Nevertheless, other circuits may also be involved. Firstly, several studies indicated that NAc projection neurons did not directly innervate melanin-concentrating hormone (MCH)- or orexin-containing neurons[14,44]. Besides, scientists suggested NAc neurons may indirectly send signals to MCH neurons via glutamatergic neurons in the anterior LH[44]. Thus, stimulation of inhibitory NAc $D_1R$ to LH pathway may inhibit sleep-promoting MCH neurons and induce wakefulness. Furthermore, LH GABA neurons can fall into separate and distinct populations which project to different brain areas involved in sleep–wake regulation[30,31]. Finally, these neurons also contain interneurons, which inhibit other neurons in the LH by a local network[45].

Recently, Liu et al.[46] identified a GABAergic subpopulation of neurons (lhx6+) in the ventral zona incerta (ZI), which is adjacent to the LH, that promotes sleep. They also conducted a retrograde tracing study and confirmed a monosynaptic projection from NAc to ZI lhx6+ neurons. We examined IPSCs in ZI neurons following optogenetic stimulation of NAc $D_1R$ neuron terminals in brain slices. Indeed, we observed sparse ChR2-positive fibers around lhx6+ neurons in the ZI (Supplementary Fig. 9a, b), but we did not detect functional connection ($n = 7$ cells from 2 mice), suggesting that NAc $D_1R$ – ZI pathway might not be crucial for arousal regulation of NAc $D_1R$ neurons. Taken together, NAc $D_1R$ – LH circuit is independent and crucial in the modulation of arousal, but detailed functional circuit mapping deserves further study.

It has been reported that NAc $A_{2A}R$ – VP pathway is essential for mediating sleep[23], and although both NAc $D_1R$- and $D_2R$-expressing neurons innervate the VP (Supplementary Fig. 7c–e), quantitative analysis revealed less than 3% colocalization of $D_1R$- and $D_2R$-expressing fibers in the VP, strongly indicating separate $D_1R$- and $D_2R$-neurons projections to the VP[47]. Whether and how the NAc $D_1R$ – VP pathway is mediating wakefulness is unknown, and since the VP serves as an output nucleus of the basal ganglia that sends inhibitory projections to different brain regions[47,48], it raises the possibility that the NAc $D_1R$ – VP and NAc $D_2R$ – VP pathways control sleep–wake states by targeting different subsets of VP neurons. Further studies are needed to uncover the role of the NAc $D_1R$ – VP pathway in sleep–wake regulation.

Patients or animals with acute morphine administration display insomnia[7,21], and an immunohistological study indicates that acute morphine injection predominantly activates NAc $D_1R$ neurons[22], which is consistent with our findings that activation of $D_1R$ neurons induces wakefulness. Therefore, our data support a role of NAc $D_1R$ neurons in pathological sleep disorders.

In conclusion, we found that NAc $D_1R$-expressing neurons are essential in controlling wakefulness and are involved in physiological arousal via the LH and midbrain circuits, suggesting that the NAc should be considered as a potential target area for therapy in neuropsychiatric disorders with sleep–wake alterations.

## Methods

**Animals**. $D_1R$-Cre ((B6.FVB(Cg)-Tg(Drd1a-Cre) EY266Gsat/Mmucd, GENSAT) mice[49] were kindly provided by Jiang-Fan Chen. $D_2R$-Cre (B6.FVB(Cg)-Tg(Drd2-cre)ER44Gsat/Mmucd, GENSAT) mice were generously provided by Dong-Min Yin. GAD67-GFP knock-in mice[50] were obtained from Yuchio Yanagawa. We crossed $D_1R$-Cre and GAD67-GFP mice to generate a new mouse $D_1R$-Cre:: GAD67-GFP. Transgenic mice had been backcrossed in the C57BL/6 line for a minimum of four generations. Male mice at 10–16 weeks of age were used for behavioral experiments, whereas mice of either sex at 4–8 weeks of age were used for in vitro patch-clamp experiments. Mice had access to food and water ad libitum and were maintained at constant temperature (22–24 °C), humidity (40–60%), and 12 h light/dark cycle (100 Lux, light on at 07:00)[51]. All experimental procedures involving animals were approved by the Animal Experiment and Use Committee at the Shanghai Medical School of Fudan University.

**AAV generation**. The AAVs of serotype rh10 for AAV-hSyn-DIO-hM3Dq-mCherry, AAV-hSyn-DIO-hM4Di-mCherry, AAV-hSyn-DIO-ChR2-mCherry, and AAV-hSyn-DIO-mCherry were generated by tripartite transfection (AAV2/10 expression plasmid, adenovirus helper plasmid, and pAAV plasmid) into 293A cells. After 3 days, the 293A cells were resuspended in artificial cerebrospinal fluid (aCSF), freeze–thawed four times, and treated with benzonase nuclease (Millipore) to degrade all forms of DNA and RNA. Subsequently, the cell debris was removed by centrifugation and the virus titer in the supernatant was determined using an AAVpro Titration Kit for Real-Time PCR (Takara). The final viral concentrations of the transgenes were $1–2 \times 10^{12}$ genome copies/mL. AAV vector carrying the EF1α-DIO-GCaMP6f construct was packaged into AAV2/9 serotype with titers $1–2 \times 10^{12}$ genome copies/mL (Shanghi Taiting Biological Co., Ltd. Shanghai, China).

**Fig. 7** Chemogenetic inhibition of NAc $D_1R$ neurons increases NREM sleep and promotes nest-building. **a** Drawings of superimposed AAV injection sites in the NAc of $D_1R$-Cre mice ($n = 9$, indicated with different colors). **b** Representative traces showing that CNO decreased the number of evoked action potentials in an hM4Di-expressing neuron. **c** Time course of wakefulness, NREM sleep, and REM sleep after administration of vehicle or CNO at 09:00 ($n = 9$, repeated-measures ANOVA; $F_{1,16} = 26.7$ (wake), 24.9 (NREM), 0.3 (REM); $P = 8 \times 10^{-5}$ (wake), $1 \times 10^{-4}$ (NREM), 0.6 (REM)). **d** Total amounts of each stage for 3 h following treatments during the light period ($n = 9$, paired $t$ test; $t_8 = 7.2$, $P = 9 \times 10^{-5}$ (wake); $t_8 = 6.2$, $P = 3 \times 10^{-4}$ (NREM); $t_8 = 0.5$, $P = 0.6$ (REM)). **e** EEG power density of NREM and REM sleep after CNO and vehicle injections. Red lines represent $P < 0.05$ ($n = 9$, paired $t$ test). **f** Total amounts of each stage for 3 h following treatments during the dark period ($n = 9$, paired $t$ test; $t_8 = 2.9$, $P = 0.02$ (wake); $t_8 = 2.9$, $P = 0.02$ (NREM); $t_8 = 1.9$, $P = 0.09$ (REM)). **g** EEG power density of NREM and REM sleep after CNO and vehicle injections ($n = 9$, $P > 0.05$, paired $t$ test). **h** Representative pictures of hM4Di mouse cages following vehicle and CNO administrations at the end of the 3 h test. **i** Nesting score during the 3 h test (1, poor; 5, good). Wilcoxon matched-pairs signed rank test, mCherry: $n = 8$, $Z = 0.4$, $P = 0.7$; hM4Di: $n = 8$, $Z = 2.0$, $P = 0.047$. **j** Inhibition of NAc $D_1R$ neurons did not affect food intake ($n = 8$ for hM4Di group and $n = 6$ for control group, paired $t$ test; 2 h, mCherry: $P = 0.8$, hM4Di: $P = 0.2$; 24 h, mCherry: $P = 0.5$, hM4Di: $P = 0.7$). Data represent mean ± s.e.m. *$P < 0.05$, **$P < 0.01$, NS not significant

**Stereotaxic surgery.** All mice used were anesthetized with pentobarbital sodium (50 mg/kg, i.p.) for surgical procedures and placed into a stereotactic frame (RWD, Shenzhen, China). A burr hole was made, and a fine glass pipette (15–20 µm tip) containing recombinant AAV-DREADD, AAV-ChR2, or AAV-mCherry virus was inserted bilaterally into the NAc (anteroposterior (AP) = + 1.4 mm, mediolateral (ML) = ± 1.0 mm, dorsoventral (DV) = −3.8 mm). A total of 70–100 nL virus was delivered to each site over a 10 min period via nitrogen gas pulses of 20–40 psi using an air compression system, and the needle was left in place for at least 5 min to permit diffusion as previously described[52]. Mice that received bilateral injections were used for all experiments and received additional surgical implants after viral injection as described below. Only data from mice in which the infection was confirmed were accepted.

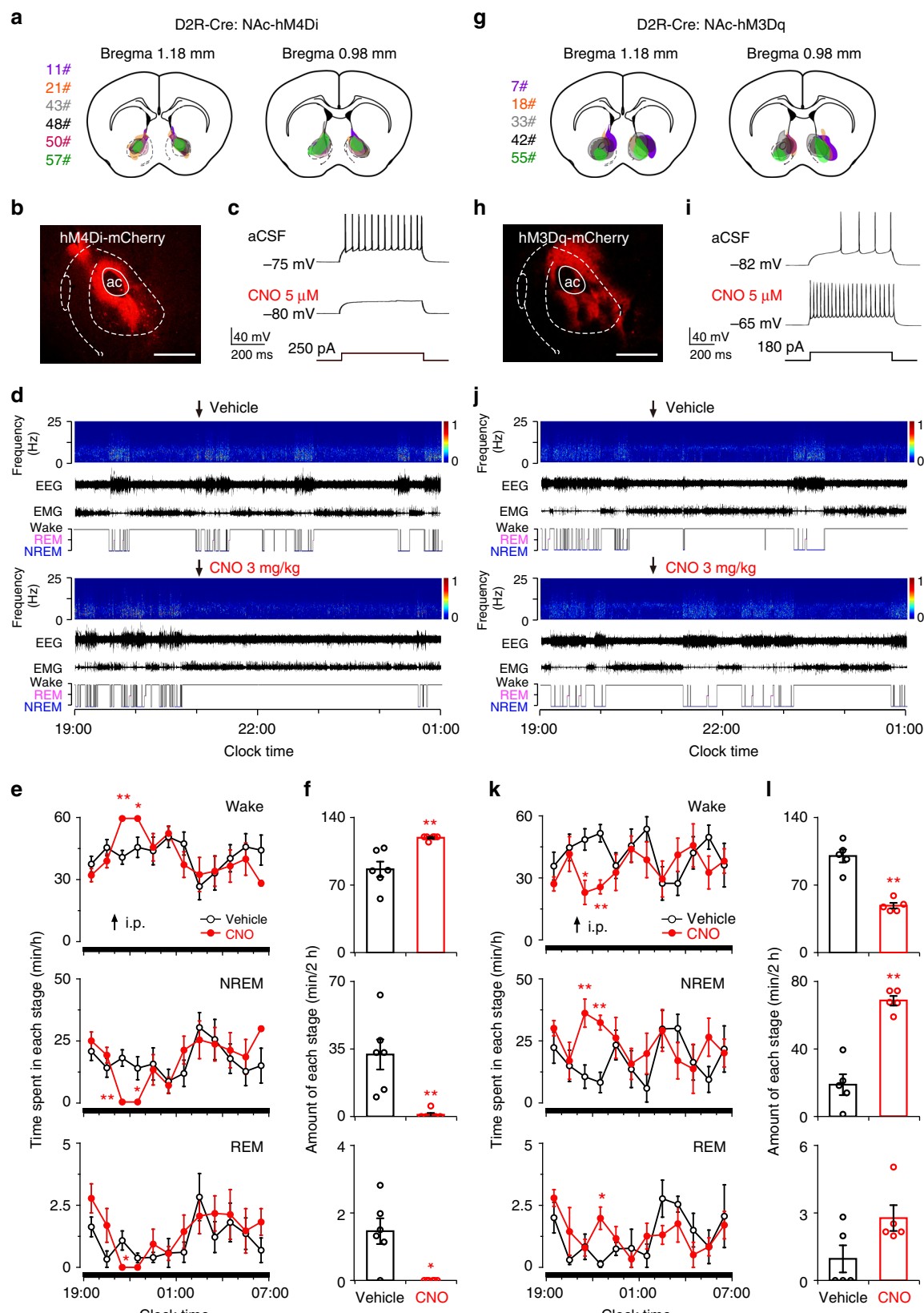

At 14 days after AAV injection, mice were implanted with EEG and EMG electrodes for polysomnographic recordings. EEG and EMG signals were recorded from stainless steel screws inserted on the skull and 2 flexible silver wires inserted in the neck muscle, respectively. For in vivo optogenetic experiments, additional stainless steel guide cannulas (8 mm length, RWD, Shenzhen, China) were positioned bilaterally above the NAc, the LH (AP − 1.5 mm, ML ± 1.0 mm, DV − 4.0 mm), or the midbrain (substantia nigra pars compacta/ventral tegmental area (SNc/VTA); AP − 3.0 mm, ML ± 1.0 mm, DV − 3.8 mm) before implanting EEG/EMG electrodes. Cannulas and EEG/EMG electrodes were affixed to the skull with dental cement. The scalp wound was sutured, and mice were allowed to recover in a warm plate until it resumed normal activity.

**Neuronal tracing.** $D_1R$-Cre mice were injected with Alexa Fluor 488-conjugated cholera-toxin subunit (CTB-488, 0.1%, Life Technologies, USA) either into the DMS (300 nL, AP + 0.7 mm, ML −1.6 mm, DV −2.4 mm) or the NAc (200 nL, AP + 1.4 mm, ML − 1.0 mm, DV −3.8 mm) 2–3 weeks after AAV-ChR2 injection. Then, 7 days later, brains were processed for fluorescence imaging or prepared for in vitro electrophysiological studies.

**Polygraphic recordings and analysis.** After 2–3 weeks for postoperative recovery and transgene expression, animals were housed individually in recording chambers and connected to the EEG/EMG headstages. The recording cable was attached to a slip-ring unit so that the movement of mice would not be restricted. Mice were habituated to recording cables for 3–4 days before starting recording.

EEG/EMG signal was recorded under baseline (free moving) and different treatment conditions (chemogenetic or optogenetic stimulations) over several days. For DREADD experiments, all mice received vehicle or CNO (3 mg/kg, C2041, LKT) treatment on 2 consecutive days at 09:00 (inactive period) or 21:00 (active period), separated by a 3-day wash-out period. After recording, mice were killed 2 h after vehicle or CNO treatment and then used for immunohistochemical staining.

EEG/EMG signal was amplified, bandpass filtered (EEG, 5–30 Hz; EMG, 40–200 Hz), digitized at 128 Hz, and recorded with Vital Recorder software (Kissei Comtec). Sleep state was scored using sleep analysis software (SleepSign, Kissei Comtec). All scoring was automatic based on EEG and EMG waveforms in 10 s epochs for chemogenetic studies, while the sleep–wake data were divided into 4 s epochs in optogenetic and fiber photometry studies. We defined wakefulness as desynchronized EEG and high levels of EMG activity, NREM sleep as synchronized, high-amplitude, low-frequency (0.5–4 Hz) EEG signals in the absence of motor activity, and REM sleep as having pronounced theta like (4–9 Hz) EEG activity and muscle atonia. Vigilance states assigned by SleepSign (Kissei Comtec, Nagano, Japan) were examined visually and corrected manually if necessary. Latency to NREM sleep was defined as the time from the end of the injection to the onset of the first NREM sleep episode which lasts more than 20 s.

**Optogenetic stimulation during polygraphic recordings.** Optical fiber cables (1 m long, 200 μm diameter; ThorLabs) were placed inside the implanted cannulas 1 day before stimulation. A rotating optical joint (FRJ_FC-FC, Doric Lenses, Canada) was used to relieve torque and was attached to the external end of the optical fiber (FC connection). The joint was connected via a fiber to a 473 nm blue laser diode (Newton Inc., Hangzhou, China). Light pulse trains were generated via a stimulator (SEN-7103, Nihon Kohden, Japan) and output through an isolator (ss-102J, Nihon Kohden). For acute photostimulation, each stimulation epoch was applied 16 s after identifying a stable NREM or REM sleep event by real-time online EEG/EMG analysis. Light pulse trains (5 ms pulses of various frequencies and durations) were programmed and conducted during the inactive period. For chronic photostimulation procedure, programmed light pulse trains (5 ms pulses at 20 Hz for 10 s and at 30 s intervals for 1 h) were used. Light stimulation was done from 09:00 to 10:00. EEG/EMG recorded during the same period on the previous day served as baseline control. The sleep–wake cycle parameters (amount, bouts, and mean duration of wakefulness, REM sleep, and NREM sleep, as well as sleep–wake transitions) were quantified by offline scoring of the entire hour for

each animal. Mice that received chronic photostimulation were killed 30 min after the final stimulation for c-Fos staining. Power intensities of blue or yellow light at the tip of the optical fiber were measured by a power meter (PM10, Coherent) and calibrated to emit 20–30 mW/mm².

**EEG power spectral analysis.** Signals were digitally filtered and spectrally analyzed by fast Fourier transformation using SleepSign software. EEG power spectra were computed for 8–12 photostimulated events (16 s per event) per mouse, with 16 s before photostimulation as baseline. Mean spectral density of all stimulated events per animal was sorted into successive 0.25 Hz frequency bands between 0 and 25 Hz. Then, each frequency band was normalized to the sum of the power over the entire range (0–25 Hz).

**Fiber photometry.** Following AAV-EF1α-DIO-GCaMP6f virus injection, an optical fiber (125 μm O.D., 0.37 numerical aperture (NA); Newdoon, Shanghai) was placed in a ceramic ferrule and inserted towards the NAc. Fiber photometry[53] uses the same fiber to both excite and record from GCaMP in real time. To record fluorescence signals, laser beam was passed through a 488 nm laser (OBIS 488LS; Coherent), reflected off a dichroic mirror (MD498; Thorlabs), focused by objective lens (Olympus), and coupled through a fiber collimation package (F240FC-A, Thorlabs) into a patch cable connected to the ferrule of the upright optic fiber implanted in the mouse via a ceramic sleeve (125 μm O.D.; Newdoon, Shanghai). GCaMP6 fluorescence was bandpass filtered (MF525–39, Thorlabs) and collected by a photomultiplier tube (R3896, Hamamatsu). An amplifier (C7319, Hamamatsu) was used to convert the photomultiplier tube current output to voltage signals, which was further filtered through a low-pass filter (40 Hz cut-off; Brownlee 440). The photometry voltage traces were downsampled using interpolation to match the EEG/EMG sampling rate of 512 Hz by using a Power1401 digitizer and Spike2 software (CED, Cambridge, UK). Analysis of the resulting signal was performed with custom written MATLAB software.

**Fiber photometry data analysis.** Photometry data were exported to MATLAB Mat files from Spike2 for further analysis. We derived the value of the photometry signal ($\Delta F/F$) by calculating $(F − F_0) / F_0$, where $F_0$ is the baseline fluorescence signal. For the sleep–wake analysis, we recorded data for 4–6 h per mouse, and calculated the averaged $\Delta F / F$ during all times of vigilance states. For analyzing the state transition, we determined each state transition and aligned $\Delta F/F$ in a ±50 s window around that point was calculated.

**Nest-building behavior.** We injected hM3Dq and hM4Di mice intraperitoneally with either vehicle or 3 mg/kg CNO during the light (09:00) and dark period (19:00), respectively, and they were placed in the new home cage with 2 × 6 pieces (50 × 70 mm) of paper towel (Qingfeng, China). We tested nest-building 3 h later. Each animal was evaluated following both vehicle and CNO injection, with at least 4 days of wash-out. The nests were assessed according to the 5-point scale from 1 to 5 as follows[54]: 1 = paper towels not noticeably touched, 2 = paper towels scattered throughout the cage, 3 = paper towels with little damage in one corner of the cage, 4 = partially shredded but often no identifiable nest, 5 = mostly shredded into small pieces and arranged into a circular nest. In all these experiments, all nests were assessed by two raters blind to the treatments to avoid any possible biases in scoring the nest completeness.

**Feeding behavior assays.** During feeding behavior experiments, mice were individually housed in home cages. All animals were mildly food restricted by removing chow food during the final 8 h before light off (19:00). All tests were performed using a counter-balanced within-subjects design. Mice received i.p. injections of either vehicle or CNO (3 mg/kg) at 19:00, 15 min before access to ~5 g of fresh standard chow. Food intake was manually calculated in the home cage by briefly removing the food from the hopper and obtaining its weight during the dark period. Food intake was measured at 2 h and 24 h after injection. Subsequent

**Fig. 8** Chemogenetic inhibition of NAc $D_2R$ neurons increases wakefulness, while activation induces NREM sleep in $D_2R$-Cre mice. **a** Drawings of superimposed AAV-DIO-hM4Di injection sites in the NAc of $D_2R$-Cre mice ($n = 6$, indicated with different colors). **b** Representative image of hM4Di-mCherry expression in the NAc. Scale bar: 500 μm. **c** CNO applied to the bath reduced firing rate in response to 250 pA current injection in an hM4Di-expressing $D_2R$ neuron of brain slice. **d** Typical examples of compressed spectral array EEG, EMG, and hypnograms over 6 h following administration of vehicle or CNO in a $D_2R$-Cre mouse with bilateral hM4Di receptor expression in the NAc. **e** Time course of wakefulness, NREM sleep, and REM sleep following injection of vehicle or CNO in mice expressing hM4Di receptor in NAc $D_2R$ neurons ($n = 6$, repeated-measures ANOVA; $F_{1, 10} = 16.8$ (wake), 15.9 (NREM), 14.6 (REM); $P = 0.002$ (wake), 0.003 (NREM), 0.003 (REM)). **f** Total time spent in each stage for 2 h after vehicle or CNO injection ($n = 6$, paired $t$ test; $P = 0.009$ (wake), $P = 0.01$ (NREM), $P = 0.012$ (REM)). **g** Drawings of superimposed AAV-DIO-hM3Dq injection sites in the NAc of $D_2R$-Cre mice ($n = 5$). **h** Representative image of hM3Dq-mCherry expression in the NAc. Scale bar: 500 μm. **i** CNO applied to the bath increased firing rate in response to 180 pA current injection in an hM3Dq-expressing $D_2R$ neuron of brain slice. **j** Typical examples of compressed spectral array EEG, EMG, and hypnograms over 6 h following administration of vehicle or CNO in a $D_2R$-Cre mouse with bilateral hM3Dq receptor expression in the NAc. **k** Time course of wakefulness, NREM sleep, and REM sleep after administration of vehicle or CNO to mice expressing hM3Dq in NAc $D_2R$ neurons ($n = 5$, repeated-measures ANOVA; $F_{1,8} = 17.8$ (wake), 16.9 (NREM), 7.6 (REM); $P = 0.003$ (wake), 0.003 (NREM), 0.025 (REM)). **l** Total time spent in each stage for 2 h after vehicle or CNO injection ($n = 5$, paired $t$ test; $P = 0.003$ (wake), $P = 0.002$ (NREM), $P = 0.1$ (REM)). *$P < 0.05$, **$P < 0.01$

injections were separated by at least 24 h. Food restriction continued during wash-out days.

**Locomotor activity recordings**. Locomotor activity for hM3Dq experiments in $D_1R$-Cre mice was assessed according to previous studies[55]. Briefly, movement of an individual mouse in a recording cage (28 cm × 16.5 cm × 13 cm) was detected with an overhead infrared sensor (Biotex, Kyoto, Japan). Each mouse was habituated to the recording cage for 24 h and recorded for the following 3 days. Mice were injected with vehicle or CNO (1 mg/kg or 3 mg/kg) at 09:00 or at 21:00 .

**In vitro electrophysiology**. At 3 to 4 weeks after AAV-ChR2 injections, $D_1R$-Cre mice or $D_1R$-Cre::GAD67-GFP mice were anesthetized and perfused transcardially with ice-cold modified aCSF saturated with 95% $O_2$ and 5% $CO_2$ and containing (in mM): 215 sucrose, 26 $NaHCO_3$, 10 glucose, 3 $MgSO_4$, 2.5 KCl, 1.25 $NaH_2PO_4$, 0.6 mM Na-pyruvate, 0.4 ascorbic acid, and 0.1 $CaCl_2$. Brains were then rapidly removed, and acute coronal slices (300 μm) containing the NAc, lateral hypothalamus, or midbrain were cut on a vibratome (VT1200, Leica, Germany) in ice-cold modified aCSF. Next, slices were transferred to a holding chamber containing normal recording aCSF (in mM): 125 NaCl, 26 $NaHCO_3$, 25 glucose, 2.5 KCl, 2 $CaCl_2$, 1.25 $NaH_2PO_4$ and 1.0 $MgSO_4$, and allowed to recover for 30 min at 32 °C. Then, slices were maintained at room temperature (RT) for 30 min before recording.

During recording, slices were submerged in a recording chamber superfused with aCSF (2 mL/min) at 30–32 °C. Slices were visualized using a fixed-stage upright microscope (BX51W1, Olympus, Japan) equipped with a 40× water immersion objective and an infrared-sensitive CCD camera. Expression of ChR2 was confirmed by visualization of mCherry fluorescence in $D_1R$-expressing neurons and axon terminals. In $D_1R$-Cre::GAD67-GFP mice, GAD67-GFP neurons were identified based on their GFP expression. Recordings were performed in regions with bright mCherry fluorescence. Patch pipettes were fabricated from thick-walled borosilicate glass capillaries (1.5 mm outer diameter, 0.86 mm internal diameter, Vital Sense; Scientific Instruments Co., Ltd, China) using a 2-step vertical puller (Narishige, PC-10) and had resistances between 4 and 6 MΩ. Recording pipettes were filled with an internal solution containing (in mM): 105 potassium gluconate, 30 KCl, 10 phosphocreatine, 4 ATP-Mg, 0.3 EGTA, 0.3 GTP-Na, and 10 HEPES (pH 7.3, 285–300 mOsm). In some experiments, 0.1% biocytin (vol/vol, Sigma) was included in the internal solution. Recordings were conducted in the whole-cell or cell-attached configuration using a Multiclamp 700B amplifier (Axon Instruments). Signals were filtered at 4 kHz and digitized at 10 kHz with a DigiData 1440A (Axon Instruments). Data were acquired and analyzed with pClamp10.3 software (Axon Instruments).

Responses were evoked by 5 ms light flashes (473 nm, 1–100 Hz) delivered from a microscope-mounted blue LED (Lumen Dynamics, Canada) through the objective lens directed onto the slice. The power of the LED light was 3–5 mW. In the voltage-clamp mode, cells were held at −70 mV. When needed, 25 μM d-(−)-2-amino-5-phosphonopentanoic acid (d-APV), 5 μM 6-cyano-7-nitroquinoxaline-2,3-dione (CNQX), and 100 μM picrotoxin (PTX) were added to block NMDA, AMPA, and $GABA_A$ receptors, respectively. Series resistance ($R_s$) compensation was not used. Therefore, cells with $R_s$ changes over 20% were discarded.

**Single-cell RT-PCR**. At the end of the recording, the cytosolic content was aspirated into the patch pipette, and expelled into a 200 μL PCR tube (Axygen, USA) as described previously[56]. The single-cell RT-PCR protocol was designed to detect the presence of mRNAs coding for Vgat. Reverse transcription and the first round of PCR amplification were performed with gene-specific multiplex primer using the SuperScript III One-Step RT-PCR kit (12574018, ThermoFisher, USA). The first reaction was performed as follows: 30 min at 55 °C, 2 min at 94 °C; 35 cycles of 15 s at 94 °C, 30 s at 55 °C, and 50 s at 68 °C; and 5 min at 68 °C. A second PCR was then conducted with nested primers. The first PCR product (1–2 μL) was used as template for a second PCR. In this second round, the reaction was performed as follows: 2 min at 94 °C; 35 cycles of 15 s at 94 °C, 30 s at 55 °C, and 30 s at 68 °C; and 5 min at 68 °C. The final PCR products were visualized by electrophoresis in agarose gels (2.0%) with SafeGel. The expected size of each final PCR product is VGAT 250 bp. The specific primers for Vgat gene were custom designed and synthesized (Sangon, Shanghai) based on a previous reference[57]. Sense, 5′ATT-CAGGGCATGTTCGTGCT3′; antisense, 5′ ATGTGTGTCCAGTTCATCAT3′; nested, 5′TGATCTGGGCCACATTGACC3′.

**Immunohistochemistry**. After whole-cell recording, slices containing biocytin-loaded cells were fixed in 4% paraformaldehyde (PFA), nonspecific binding was blocked with 5% donkey serum in phosphate-buffered saline (PBS). Then, slices were incubated overnight at 4 °C in PBS containing 0.3% Triton-X (PBST), primary antibodies, and streptavidin conjugated to Alexa 405 or Alexa 488 (1:1000; Invitrogen Molecular Probes, USA). Primary antibody was raised against TH (1:3000; rabbit host; Millipore, USA). The following day, slices were washed 3 times in PBS and incubated in PBST containing secondary antibodies for 12 h at 4 °C. Secondary antibody was donkey anti-rabbit 488 or 647 (1:500, Jackson ImmunoResearch).

Finally, slices were washed in PBS and mounted on glass slides using Fluoromount-GTM (Southern Biotech, 0100–01).

For double immunostaining of c-Fos and mCherry, after optical stimulation or CNO administration, mice were deeply anesthetized by chloral hydrate (400 mg/kg) and then perfused intracardially with 20 mL PBS followed by 40 mL 4% PFA. Brains were removed, postfixed for 6 h in 4% PFA, and then incubated in 20% sucrose phosphate buffer (PB) at 4 °C until they sank. Coronal sections (30 μm) were cut on a freezing microtome (CM1950, Leica, Germany) in 4 series. The floating sections were washed in PBS and incubated with a rabbit polyclonal antibody against c-Fos (1:10000, pc-38, Calbiochem) in PBST for 48 h at 4 °C on an agitator. After washing, sections were incubated with a biotinylated goat anti-rabbit IgG antibody (1:1000, BA-1000, Vector Laboratories) followed by incubation in an avidin-biotin peroxidase complex (ABC) solution (1:1000, PK-6100, Vector Laboratories) for 1 h. After rinsing, the sections were immersed in a 3,3-diaminobenzidine-4 HCl (DAB) and nickel solution (SK-4100, Vector Laboratories) for 5–10 min at RT, in which Fos-immunoreactive neurons were identified by the presence of black reaction product. The following day, the c-Fos immunostained sections were incubated in a rabbit polyclonal antibody against DsRed (mCherry tag, 1:5000, 632496, Clontech) overnight at 4 °C. Amplification steps were similar to those described above, except that the last step was performed in a DAB solution without nickel ammonium sulfate. Finally, the sections were mounted on glass slides, dried, dehydrated, and coverslipped.

For circuit mapping, mouse brains were fixed and prepared as described above. Coronal cryostat sections (30 μm) were incubated in 3% normal donkey serum PBST for 2 h and then in primary antibodies in PBST overnight at RT. The following primary antibodies were used: rabbit anti-SP (1:400, ABN1369, Millipore), goat anti-Orexin-A (1:500, sc-8070, Santa Cruz Biotechnology), mouse anti-TH (1:1000, T2928, Sigma), goat anti-PV (1:3000, PVG213, Swant), and mouse anti-lhx6 (1:500, sc-271433, Santa Cruz Biotechnology). Next, sections were rinsed 3 times in PBS and incubated for 2 h at RT with the following secondary antibodies: donkey anti-rabbit 488 (for SP, 1:500, 711-545-152, Jackson ImmunoResearch), donkey anti-mouse 488 (for TH and lhx6, 1:1000, A-21202, Invitrogen), donkey anti-goat 647 (for Orexin-A, 1:1000, A21447, Invitrogen), and donkey anti-goat 488 (for PV, 1:500, 711-545-147, Jackson ImmunoResearch). Some sections were also incubated in 4,6-diamidino-2-phenylindole (DAPI, 1:3000, D1306, Invitrogen) for 7 min at RT. Finally, sections were washed 2 times in PBS, mounted on slides, and coverslipped.

Nonconfocal images were captured by a fluorescence microscope (IX71, Olympus). High-resolution fluorescence images were collected on a confocal microscope (FV-1000, Olympus). Digital images were processed using FV-1000 Viewer 2.0 and Adobe Photoshop CS3 software to minimally adjust brightness and contrast. All digital images were processed in the same way between experimental conditions to avoid artificial manipulation between different data sets.

**Electron microscopy**. Under deep anesthesia, $D_1R$-Cre mice injected with AAV-ChR2-mCherry in the NAc were briefly rinsed transcardially with 6–8 mL saline and then perfused with 100 mL ice-cold fixative which contains 4% PFA, 15% saturated picric acid in 0.1 M PB, and 0.5% glutaraldehyde. Immediately after perfusion–fixation, the brain was removed from the skulls and postfixed in 4% PFA for 2 h at 4 °C. The brain containing the midbrain was cut into coronal sections (40 μm) with a vibratome (VT1000S, Leica, Germany) and collected in 0.05 M PB for subsequent mCherry and TH double staining. The sections were placed in 0.05 M PB (pH 7.4) containing 10% (v/v) glycerol and 25% (w/v) sucrose for 1 h and then freeze–thawed with liquid nitrogen to enhance penetration of antibody. The sections were then incubated in rabbit polyclonal antibody against DsRed (1:5000) in 0.05 M PB at 4 °C for 30–36 h. At RT, the sections were then incubated with goat anti-rabbit biotinylated IgG (1:1000) for 3 h, followed by incubation in ABC complex for 3 h. After mCherry immunoreactivity was visualized with DAB following the same procedure as we used for light microscopy, the brain sections were incubated in rabbit anti-TH antibody (1:1000, Millipore) containing 5% normal donkey serum at 4 °C for 30–36 h. Then, the sections were incubated in goat anti-rabbit biotinylated IgG (1:1000) followed by ABC solution at RT for 3 h. A chromogen of the VIP substrate kit (SK-4600, Vector Laboratories) was used to visualize TH immunoreactivity. The double-labeled sections were osmicated with $OsO_4$ (2%), dehydrated in a graded series of ethanol, and embedded flat in Epon 12 (Ted Pella, USA) using embedding capsules (TAAB, UK). The sections embedded were observed under a light microscope, and the area containing labeled axons and TH cell bodies in the midbrain was sampled. Ultrathin sections were obtained using an ultramicrotome at 70 nm, stained with uranyl acetate and lead citrate, and then examined in a CM-120 transmission electron microscope (Philips, Netherlands).

**Statistical analysis**. Data are presented as mean ± s.e.m. Sample sizes were chosen based on previous studies[12,58]. No method of randomization or blinding was used in the experiments except for nest-building test. Two-way repeated-measures analysis of variance (ANOVA) was used to perform group comparisons with multiple measurements. Paired and unpaired $t$ tests were used for single value comparisons. One-way ANOVA was used to compare more than two groups, followed by post hoc Turkey test analysis for multiple comparisons. Two sets of frequencies were analyzed by chi-square test. Nest-building behavior was tested by

Wilcoxon matched-pairs signed rank tests. A two-tailed $P$ value of <0.05 was considered statistically significant. Statistical analysis was performed using SPSS 16.0 software.

**Data availability**. The data that support the findings of this study are available from the corresponding author on request.

**Code availability**. Data analyses were conducted in Matlab using scripts available from the corresponding author upon reasonable request.

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

## Acknowledgements

We are grateful to H. Dong, Z.-L. Liu, R.-F. Wang, and T. Fang for technical assistance, Dr. D.-M. Yin for providing D$_2$R-Cre mice, Dr. Y. Yanagawa for providing GAD67-GFP knock-in mice, and Dr. Z.-G. Yang for valuable discussion. This study was supported in part by grants-in-aid for scientific research from the National Natural Science Foundation of China (81420108015, 31530035, 31671099, 31471064, 31571103, 31421091, 81630040, and 31771178); the National Basic Research Program of China (2015CB856401); and the Shanghai Outstanding Academic Leaders Plan (Z. L. Huang).

## Author contributions

Y.-J.L., L.W., W.-M.Q., and Z.-L.H. designed the experiments. Y.-J.L. and L.W. performed patch-clamp electrophysiology. Y.-J.L. and Y.-D.L. performed behavioral tests and analyzed the fiber photometry data. Y.-D.L., S.-R.Y., and J.W. performed histology and immunostaining. X.-S.Y. performed immunoelectron experiment. M.L. and Y.C. provided the viral vectors. Y.-J.L., Y.-D.L., and W.-M.Q. collected and analyzed the data. Y.-J.L., Y.-D.L., L.W., J.-F.C., and Z.-L.H. wrote the manuscript, and all of the authors helped with the revision of the manuscript.

## Additional information

**Competing interests:** The authors declare no competing interests.

