## [Peer Review File · Nature Communications]

Reviewers' comments:

Reviewer #1 (Remarks to the Author):

This is a solid manuscript by Luo, Li, Wang et al. Utilizing state of the art tools the authors show that acute activation of nucleus accumbens NAc D1Rs promote transitions to wakefulness (they initiate it), while chronic opto activation of D1R neurons promotes prolonged wakefulness. Circuit mapping shows that these NAc D1R MSNs preferentially target non-DA neurons (probably GABA) in the VTA/SN, however they find no direct input from D1R MSNs to SN DA neurons, very sparse input to VTA DA neurons. Overall this is a well executed series of studies. The overall novelty of their work was diminished somewhat by a similar paper recently published by the Delecea group, however, it is still an important addition to literature. My suggestions and concerns are detailed below.

Thoughts:

- Firstly, as mentioned above, the subject is largely lacking in novelty aside from the D1R output portion of the results (they have essentially just focused on the post-synaptic aspect of the previous DeLecea study). However, the cell-type specific circuit mapping of D1R NAc MSNs-VTA neurons is valuable in elucidating the connectivity mediating the behavioral effect observed.
- The focus on D1R MSNs in the NAc should really be supplemented by complementary studies in D2R MSNs. It doesn't seem appropriate to just refer to other studies for this.
- Overall, the study would be strengthened by focusing on the NAc-VTA/SN component of the story and leaving out the LH. The LH part seems like it was tossed in and thus seems incomplete.
 - o Not to mention that the LH data is not in agreement with previous studies→ another reason why they need to either perform detailed circuit analysis or just leave it out.

Minor notes:

- o Fig 1:
 - For FP experiments, they used 3 mice but instead of averaging the responses within mice and reporting that, they report all data points together for each condition→ this doesn't seem like how this should be done
- o On the top of page 8, it is unclear what recordings were done here. It should be stated more clearly in the text even if it is also in the figure.
- o Why did they use male and female mice for the ephys if no female mice were used behaviorally? It's not the biggest deal but it definitely isn't ideal.
- o The laser power they reported was 20-30 mW out of the fiber tip for opto. That is very high! I would find it unlikely that there was no damage at that power level.
- o What kind of AAVs were used (serotypes)? Which promoters were used? Were they FLEX or DIO? There is not a lot of info about the viruses.

Reviewer #2 (Remarks to the Author):

In their manuscript, Luo et al. used up to date methodologies such as optogenetic, chemogenetic and calcium imaging to elegantly show that nucleus accumbens GABAergic neurons expressing dopamine D1 receptors play a crucial role in wakefulness. Overall, the demonstration is really impressive and convincing. It is also nicely described and illustrated. Below, I made some remarks and requests on the work.

Line 103-106, the authors wrote that the NAc D1R neurons start to increase their firing before wake and REM sleep at the transitions with NREM. Indeed, it is visible in Fig. 1f. It would be of interest to give a mean time value here to figure out how much the neurons anticipate states changes.

The authors state line 115 that intense Fos expression was induced in hM3Dq expressing neurons. The illustration Fig. 2C showing such expression is too small to give an idea of the number and distribution of Fos+ neurons. It would be nice to enlarge the photo or show an enlarged photo as a supplementary figure. A drawing showing the distribution of Fos with a quantification would be perfect.

Again line 163 for optogenetic, same comment on the Fos staining with a very small photography.

Line 242: are instead of were

262-263: I don't get how the authors conclude that inputs from NAc D1R neurons to nigrostriatal and mesolimbic neurons are different.

Indeed, NAc D1R neurons do not project to nigrostriatal neurons. Further, the projection of NAc on VTA neurons is not destined to DA neurons.

Line 265-284: stimulation of the ChR2 fibers in the LH and the midbrain induced waking similarly.

For the LH, the author further found that only 37% of the LH neurons were inhibited on slices. I have two questions there, is it known whether stimulating passing fibers expressing ChR2 give rise to no effect? A Fos staining might help to answer such question.

My second question would be on the types of inhibited neurons in the LH. Indeed, it should be GABAergic neurons inhibiting Wake-active neurons. The authors did look at the types of neurons responding? Either in terms of electrophysiological or neuroanatomical characteristics?

Line 318: The authors cite a publication in press showing that NAc D2R-expressing neurons strongly promote sleep. They should develop the results obtained in this study. This is indeed important in order to get the whole picture of the role of the NAc neurons in sleep.

330-331: rephrasing of the sentence is necessary

334: on the firing of DA VTA neurons, the article of Dahan et al. (2007) is missing.

340-342: repetitions again in the sentence

Line 343-344: the information on the neuronal target of adenosine in the NAc as published by Lazarus et al. is missing.

345-361: Discussion on the LH is focusing on GABAergic neurons involved in waking. However, this is relatively a new concept whereas there are many publications showing the presence of GABAergic neurons involved in sleep including a very recent one in Nature about ZI neurons. These publications need to be discussed. Missing are also publications demonstrating that a projection from the NAc to LH neurons exists. This is important since many of the fibers there are just passing in the MFB.

Line 362 They state that the NAc-VP pathway is also important for sleep without developing the arguments. Again, this is important to figure out which projection does what.

Overall, the discussion needs to provide a clearer picture on the role of the different NAc neurons and projections. They should also discuss the role of interconnections between the two populations as reported in their supplementary Fig. 5.

A drawing would be particularly helpful in that matter.

Reviewer #3 (Remarks to the Author):

The study is intensive and extensive using various cutting edge techniques, and the results appeared to be reliable. However, it was difficult to understand the functional importance of the work.

I understand from the data presented that NAc D1R neuron circuits are essential for the induction and maintenance of wakefulness, but the manuscript does not address anything about how NAc D1R neuron circuits are physiologically and pathophysiology important. The abstract is merely a list of the findings and does not address functional importance of the NAc D1R neuron circuits. Most of the experiments, except in vivo fiber photometry and electron microscopy experiments, are likely to be supraphysiological and do not provide much information regarding how these NAc D1R neuron circuits coordinate with other systems to regulate sleep and wakefulness.

Both D1R and D2R neuron exist in the NAc, and D1R and D2R-expressing neurons play complementary and sometimes opposing roles in higher brain functions. A recent research reported that direct stimulation of the NAc A2AR/D2R-expressing neurons induced remarkable non-rapid eye movement (non-REM, NREM) sleep.

These taken together with the fact that the endogenous molecule mediating the NAc D2R and D1R mediating effects is DA, it is important to study how these two DA receptive systems are regulated to control physiological sleep/wake and study how disruptions of these systems affect sleep and other behavioral states.

The authors concluded that NAc D1R-expressing neurons are essential in controlling wakefulness and are involved in physiological arousal via the LH and midbrain circuits, suggesting that the NAc should be considered as a potential target area for therapy in neuropsychiatric disorders with sleep-wake alterations.

This conclusion does not hold much in terms of functional significance and no data presented suggested that the NAc should be considered as a potential target area for therapy in neuropsychiatric disorders with sleep-wake alterations.

Response to Reviewers

Reviewer #1

This is a solid manuscript by Luo, Li, Wang et al. Utilizing state of the art tools the authors show that acute activation of nucleus accumbens NAc D1Rs promote transitions to wakefulness (they initiate it), while chronic opto activation of D1R neurons promotes prolonged wakefulness. Circuit mapping shows that these NAc D1R MSNs preferentially target non-DA neurons (probably GABA) in the VTA/SN, however they find no direct input from D1R MSNs to SN DA neurons, very sparse input to VTA DA neurons. Overall this is a well executed series of studies. The overall novelty of their work was diminished somewhat by a similar paper recently published by the Delecea group, however, it is still an important addition to literature. My suggestions and concerns are detailed below.

Thoughts:

- Firstly, as mentioned above, the subject is largely lacking in novelty aside from the D1R output portion of the results (they have essentially just focused on the post-synaptic aspect of the previous DeLecea study). However, the cell-type specific circuit mapping of D1R NAc MSNs-VTA neurons is valuable in elucidating the connectivity mediating the behavioral effect observed.

Response:

We thank the comments. As the reviewer mentioned, De Lecea found that the NAc is an important downstream nucleus for the VTA in mediating wakefulness. In our study, we provided a series of direct evidence showing that NAc D₁R-neurons control wakefulness. With the newly obtained data, we also revealed that two distinct populations of D₁R and D₂R neurons in the NAc play the opposite role in sleep-wake control. Besides, neuronal circuits mapping data showed that two important downstream targets (midbrain and LH) for NAc D₁R neurons in arousal control.

- The focus on D1R MSNs in the NAc should really be supplemented by complementary studies in D2R MSNs. It doesn't seem appropriate to just refer to other studies for this.

Response:

As requested, we carried out a complementary study in D₂R neurons by using D₂R-Cre mice and chemogenetic methods to activate or inhibit NAc D₂R-expressing neurons. The results showed that chemogenetic inhibition of NAc D₂R neurons induced wakefulness, which is similar to the result of NAc D₁R neuron activation, while activation of NAc D₂R neurons increased NREM sleep.

Based on these findings, we created a new figure (Figure 8, Pages 44-45, lines 1046-1072) and added texts to the revised manuscript (Page 12, lines 321-339). Our new results are in line with a recent paper that NAc A_{2A}R/D₂R neurons promote slow wave sleep by using adenosine A_{2A}R-Cre mice (Oishi et al., 2017), because adenosine A_{2A}Rs are co-expressed with dopamine D₂Rs in the same neurons in the NAc (Lazarus et al., 2011).

Figure 8 Chemogenetic inhibition of NAc D₂R neurons increases wakefulness, while activation induces NREM sleep in D₂R-Cre mice. (a) Drawings of superimposed AAV-DIO-hM4Di injection sites in the NAc of D₂R-Cre mice (n = 6, indicated with different colors). (b) Representative image of hM4Di-mCherry

expression in the NAc. (c) Bath applied CNO (5 μ M) reduced firing rate in response to 250 pA current injection in an hM4Di-expressing D₂R neuron of brain slice. (d) Typical examples of EEG (power spectrogram and wave traces), EMG, and hypnograms over 6 h following administration (i.p.) of vehicle or CNO in a D₂R-Cre mouse with bilateral hM4Di receptor expression in the NAc. (e) Time course changes in wakefulness, NREM sleep, and REM sleep following injection of vehicle or CNO in mice expressing hM4Di receptor in NAc D₂R neurons (n = 6, repeated-measures ANOVA; $F_{1,10} = 16.8$ (wake), 15.9 (NREM), 14.6 (REM); P = 0.002 (wake), 0.003 (NREM), 0.003 (REM)). (f) Total time spent in each stage for 2 h after vehicle or CNO injection (n = 6, paired *t* test; **P = 0.009 (wake), **P = 0.01 (NREM), *P = 0.012 (REM)). (g) Drawings of superimposed AAV-DIO-hM3Dq injection sites in the NAc of D₂R-Cre mice (n = 5). (h) Representative image of hM3Dq-mCherry expression in the NAc. (i) Bath applied CNO (5 μ M) increased firing rate in response to 180 pA current injection in an hM3Dq-expressing D₂R neuron of brain slice. (j) Typical examples of EEG (power spectrogram and wave traces), EMG, and hypnograms over 6 h following administration (i.p.) of vehicle or CNO in a D₂R-Cre mouse with bilateral hM3Dq receptor expression in the NAc. (k) Time course changes in wakefulness, NREM sleep, and REM sleep after administration of vehicle or CNO to mice expressing hM3Dq in NAc D₂R neurons (n = 5, repeated-measures ANOVA; $F_{1,8} = 17.8$ (wake), 16.9 (NREM), 7.6 (REM); P = 0.003 (wake), 0.003 (NREM), 0.025 (REM)). (l) Total time spent in each stage for 2 h after vehicle or CNO injection (n = 5, paired *t* test; **P = 0.003 (wake), **P = 0.002 (NREM), P = 0.1 (REM)).

- Overall, the study would be strengthened by focusing on the NAc-VTA/SN component of the story and leaving out the LH. The LH part seems like it was tossed in and thus seems incomplete.

o Not to mention that the LH data is not in agreement with previous studies→ another reason why they need to either perform detailed circuit analysis or just leave it out.

Response:

To better understand the NAc^{D₁R}-LH circuit in sleep-wake regulation, we added

new data to address these questions by *in vitro* electrophysiological experiment and c-Fos staining, please see supplementary figure 5 (page 51, lines 1136-1150), and texts in pages 10-11, lines 285-301.

To test whether NAc D₁R neurons directly inhibit wake-promoting LH GABA neurons (Herrera et al., 2015; Venner et al., 2016), we crossed D₁R-Cre and GAD67-GFP mice to generate a new mouse D₁R-Cre::GAD67-GFP. AAV-hSyn-DIO-ChR2-mCherry was injected into the NAc, and *in vitro* whole-cell recording of LH GABA neurons was used to simultaneously monitor light-evoked IPSCs derived from NAc D₁R neurons afferents (Supplementary Fig. 5a).

We found that NAc D₁R neurons innervated both GFP-positive and GFP-negative neurons in the LH, showing no distinct preference (67% vs. 47%; Supplementary Fig. 5b, d-g). Meanwhile, light-evoked IPSCs latency was less than 5 ms (Supplementary Fig. 5c), indicating the direct connection. Because GAD67 is a marker for GABA, our results suggested that NAc D₁R neurons would target both GABA neurons expressing GAD67 and non-GAD67 neurons including GAD65 GABAergic neurons and non-GABAergic neurons.

In vivo activation of NAc D₁R neurons by CNO induced robust c-Fos expression in LH orexin positive neurons (Supplementary Fig. 5h-i). The increased c-Fos expression may be caused by the disinhibition of LH orexin neurons. Furthermore, photostimulation of LH terminals from NAc D₁R neurons induced wakefulness. Taken together, these data indicated that the important NAc^{D₁R}→LH circuit is involved in arousal partially through disinhibiting LH wake-promoting orexin neurons.

Supplementary Figure 5. NAc D₁R neurons innervate both GAD67-GFP positive and negative LH neurons. (a) Schematic showing injection of Cre-dependent ChR2 into the NAc of D₁R-Cre mice::GAD67-GFP mice. Terminals of the infected D₁R neurons were optogenetically activated and responses were recorded in LH neurons. The picture on the right shows the location of GAD67-GFP neurons (green) in the LH with ChR2-expressing fibers from NAc D₁R neurons (red). (b) Proportion of connected and unconnected neurons that were identified based on GFP expression (n = 9 GFP (+) neurons, n = 17 GFP (-) neurons, from 3 mice, Chi-square test; $\chi^2 = 0.875$, P = 0.429). (c) Latency of light-evoked IPSCs in LH GFP (+) neurons (left axis) and GFP (-) neurons (right axis). (d-g) Typical examples of a connected biocytin-labelled neuron that was GFP (+, d, e) and GFP (-, f, g) responsive to light stimulation. (h, i) Systematic injection of CNO in D₁R-Cre-hM3Dq mice induced robust c-Fos (black) expression in LH orexin (brown) neurons. LH: lateral hypothalamus; ZI: zona incerta; f: fornix. PTX: picrotoxin.

Minor notes:

o Fig 1: • For FP experiments, they used 3 mice but instead of averaging the responses within mice and reporting that, they report all data points together for each condition→ this doesn't seem like how this should be done.

Response:

Thanks for the reviewer's question. We focused on the correlation between population activity of NAc D₁R neurons and sleep–wake state transitions based on De Lecea's paper (Eban-Rothschild et al., 2016).

o On the top of page 8, it is unclear what recordings were done here. It should be stated more clearly in the text even if it is also in the figure.

Response:

As requested, the sentence was rewritten in page 8, lines 203-207 in the revised manuscript to read: “Following AAV-hSyn-DIO-ChR2-mCherry virus injections into the NAc of D₁R-Cre mice (Supplementary Fig. 7a, b), we observed projections to the ventral midbrain. NAc D₁R neuron terminals were distributed mainly in the medial substantia nigra pars compacta (SNc) and VTA, and sparsely in the substantia nigra pars reticulata (SNr) (Supplementary Fig. 7h-m)”. We also added a schematic showing neural projection sites of NAc D₁R neurons after ChR2–mCherry virus injection into the NAc of a D₁R-Cre mouse (Supplementary Fig. 7a).

Supplementary Figure 7. Neural projection sites of NAc D₁R-expressing neurons.

(a) Schema of ChR2-mCherry expressed in the NAc of D₁R-Cre mice. (b) Typical image showing the injection site of ChR2-mCherry in the unilateral NAc. (c-e) Prominent fiber terminals (red, c) are seen in the VP, was confirmed by co-staining for SP (defining borders of the VP, green, d and merged in e). (f) Low-magnification photomicrograph indicating the ChR2-mCherry axon terminals in a LH brain section immunolabeled against orexin-A (green). (g) High-magnification of boxed area in f. (h-m) Typical mCherry-positive fibers (red) in the midbrain (h, k), immunolabeling

for TH identified dopaminergic neurons (green, **i**), and mCherry/TH was observed (**j**), immunolabeling for PV identified GABAergic neurons (green, **l**), and mCherry/PV was observed (**m**). LH: lateral hypothalamus; NAc: nucleus accumbens; SNc: substantia nigra pars compacta; VP: ventral pallidum; VTA: ventral tegmental area; ac: anterior commissure; acp: anterior commissure, posterior; LV: lateral ventricle; ox: optic chiasm; SP: substance P; f: fornix; IP: interpeduncular nucleus; SNr: substantia nigra reticular part; TH: tyrosine hydroxylase; PV: parvalbumin.

o Why did they use male and female mice for the ephys if no female mice were used behaviorally? It's not the biggest deal but it definitely isn't ideal.

Response:

We used male mice for behavioral tests to avoid the influence of estrous cycles that might cause data variation. *In vitro* electrophysiological study was employed to explore the connections between NAc neurons infected with ChR2 and downstream neurons. We did not find significant differences in path-clamp data of different genders, so that we used both genders in patch-clamp study.

o The laser power they reported was 20-30 mW out of the fiber tip for opto. That is very high! I would find it unlikely that there was no damage at that power level.

Response:

Thanks for the comments. In our *in vivo* optogenetic study, we reported that light density was 20-30 mW/mm², not 20-30 mW.

Actually, the laser power from the tip of fiber (200- μ m-diameter) was lower than 3 mW. The optic fiber illuminated area was estimated as a small spot (~380 μ m diameter; 0.11 mm² area), and this area was used to calculate the light density. Thus, the light density was estimated to be lower than 25.5 mW/mm², and we reported that light density was 20-30 mW/mm². This parameter was used based on previous publications (Eban-Rothschild et al., 2016; Yuan et al., 2017).

o What kind of AAVs were used (serotypes)? Which promoters were used? Were they

FLEX or DIO? There is not a lot of info about the viruses.

Response:

The virus serotype was 10 for AAV-hSyn-DIO-hM3Dq-mCherry, AAV-hSyn-DIO-hM4Di-mCherry, AAV-hSyn-DIO-ChR2-mCherry and AAV-hSyn-DIO-mCherry, and serotype 9 for AAV-EF1 α -DIO-GCaMP6f.

As requested, we added the information in the main text (pages 4-5, lines 113-115) to read “we expressed a Cre-recombinase enabled chemogenetic excitatory system under the control of the human synapsin promoter (AAV-hSyn-DIO-hM3Dq-mCherry), via AAV injections, in the bilateral NAc of 8 D₁R-Cre mice”.

More detailed information was added to the supplemental method of the revised manuscript (page 16, lines 442-453) as below.

AAV generation. The AAVs of serotype rh10 for AAV-hSyn-DIO-hM3Dq-mCherry, AAV-hSyn-DIO-hM4Di-mCherry, AAV-hSyn-DIO-ChR2-mCherry and AAV-hSyn-DIO-mCherry were generated by tripartite transfection (AAV2/10 expression plasmid, adenovirus helper plasmid, and pAAV plasmid) into 293A cells. After 3 days, the 293A cells were resuspended in artificial cerebrospinal fluid (aCSF), freeze-thawed four times, and treated with benzonase nuclease (Millipore) to degrade all forms of DNA and RNA. Subsequently, the cell debris was removed by centrifugation and the virus titre in the supernatant was determined using an AAVpro Titration Kit for Real Time PCR (Takara). The final viral concentrations of the transgenes were $1-2 \times 10^{12}$ genome copies/mL. AAV vector carrying the EF1 α -DIO-GCaMP6f construct was packaged into AAV2/9 serotype with titres $1-2 \times 10^{12}$ genome copies/mL (Shanghi Taiting Biological Co., Ltd. Shanghai, China).”

Reviewer #2 (Remarks to the Author):

In their manuscript, Luo et al. used up to date methodologies such as optogenetic, chemogenetic and calcium imaging to elegantly show that nucleus accumbens GABAergic neurons expressing dopamine D1 receptors play a crucial role in wakefulness. Overall, the demonstration is really impressive and convincing. It is also nicely described and illustrated. Below, I made some remarks and requests on the work.

Line 103-106, the authors wrote that the NAc D1R neurons start to increase their firing before wake and REM sleep at the transitions with NREM. Indeed, it is visible in Fig. 1f. It would be of interest to give a mean time value here to figure out how much the neurons anticipate states changes

Response:

We thank the reviewer for the comments. In our study, transitions of vigilance stages were judged by sleepSign software, and the EEG/EMG raw data were divided into 4-s epochs to further analyze, whereas the calcium signal fluctuated at the millisecond level, and was sorted into successive 2-ms bands between -50 and +50 s. Thus, it might not be precise to quantify the mean time between calcium signal changes and S-W state switches.

The authors state line 115 that intense Fos expression was induced in hM3Dq expressing neurons. The illustration Fig. 2C showing such expression is too small to give an idea of the number and distribution of Fos⁺ neurons. It would be nice to enlarge the photo or show an enlarge photo as a supplementary figure. A drawing showing the distribution of Fos with a quantification would be perfect.

Response:

As requested, we enlarged the figure to clearly see the intense c-Fos expression induced by CNO in hM3Dq expressing neurons (please see Figure 2c, page 32, line 894).

Figure 2. (c) Representative images of CNO induced c-Fos (black)/mCherry (brown) colocalization in the NAc. Boxed regions in (c) are enlarged in the right panel.

Again line 163 for optogenetic, same comment on the Fos staining with a very small photography.

Response:

As requested, we enlarged the figure in revised manuscript (please see Figure 3c, page 34, line 917).

Figure 3. (b) Brain section was stained against mCherry and c-Fos to confirm that the ChR2 protein was expressed in the NAc, and arrows indicate the tip of the optical fiber above the NAc. (c) Higher magnification image of white box in b indicates abundant c-Fos immunoreactivity in ChR2-mCherry neurons following photostimulation.

Line 242: are instead of were

Response:

Thank you. We corrected.

262-263: I don't get how the authors conclude that inputs from NAc D1R neurons to nigrostriatal and mesolimbic neurons are different.

Indeed, NAc D1R neurons do not project to nigrostriatal neurons. Further, the projection of NAc on VTA neurons is not destined to DA neurons.

Response:

Sorry for not describing this point clearly. The subtitle was rewritten in page 9, lines 250-251 in the revised manuscript to read: "NAc D₁R neurons send no input to DMS-projecting SNc DA neurons, and very sparse projection to NAc-projecting VTA DA neurons".

We also rewrote the conclusion in page 10, lines 264-266 in the revised paper to read: "Taken together, our data show that NAc D1R neurons do not innervate DA neurons in the SNc, which projects to the DMS, and sparsely to DA neurons in the VTA".

Line 265-284: stimulation of the ChR2 fibers in the LH and the midbrain induced waking similarly. For the LH, the author further found that only 37% of the LH neurons were inhibited on slices. I have two question there, is it known whether stimulating passing fibers expressing ChR2 give rise to no effect? A Fos staining might help to answer such question.

Response:

We thank the important comments. In brain slices, we found that projection densities from NAc D₁R neurons are relatively sparse in the LH, compared to the midbrain. On the other hand, previous studies have shown that NAc neurons would not directly innervate MCH and orexin neurons (O'Connor et al., 2015; Sano and Yokoi, 2007), which would lead to the lower connectivity. However, we found that light induced a short latency (< 5 ms) of IPSCs in LH neurons, strongly indicating a direct innervation from NAc D₁R neurons to LH neurons.

Although stimulation of the ChR2 fibers in the LH and the midbrain induced a similar short latency of NREM sleep-to-wake transition, and increased

the amount of wakefulness, we found that behavioral performances of mice were not exactly the same, for example, mice with stimulation of terminals in the midbrain showed more obvious motor activities than NAc^{D1R}-LH pathway activation (please see supplementary movies 3 and 6). On the other hand, 2 mice with optic fiber implanted above the passing fiber in the other region were given acute and semi-chronic photostimulation that were shown weak to no effect on sleep-wake pattern (data not shown). Thus, arousal-promoting effect induced by stimulation of NAc-to-LH pathway is not due to activation of passing fiber in the LH.

As to c-Fos staining, please see next answer.

My second question would be on the types of inhibited neurons in the LH. Indeed, it should be GABAergic neurons inhibiting Wake-active neurons. The authors did look at the types of neurons responding? Either in term of electrophysiological or neuroanatomical characteristics?

Response:

To better understand the NAc^{D1R}-LH circuit in sleep-wake regulation, we obtained new data to address these questions by *in vitro* electrophysiological experiment and c-Fos staining, please see supplementary figure 5 (page 51, lines 1136-1150), and texts in pages 10-11, lines 285-301.

To test whether NAc D₁R neurons directly inhibit wake-promoting LH GABA neurons (Herrera et al., 2015; Venner et al., 2016), we crossed D₁R-Cre and GAD67-GFP mice to generate a new mouse D₁R-Cre::GAD67-GFP. AAV-hSyn-DIO-ChR2-mCherry was injected into the NAc, and *in vitro* whole-cell recording of LH GABA neurons was used to simultaneously monitor light-evoked IPSCs derived from NAc D₁R neurons afferents (Supplementary Fig. 5a).

We found that NAc D₁R neurons innervated both GFP-positive and GFP-negative neurons in the LH, showing no distinct preference (67% vs. 47%; Supplementary Fig. 5b, d-g). Meanwhile, light-evoked IPSCs latency was less

than 5 ms (Supplementary Fig. 5c), indicating the direct connection. Because GAD67 is a marker for GABA, our results suggested that NAc D₁R neurons would target both GABA neurons expressing GAD67 and non-GAD67 neurons including GAD65 GABAergic neurons and non-GABAergic neurons.

Activation of NAc D₁R neurons by CNO induced robust c-Fos expression in the LH orexin positive neurons (Supplementary Fig. 5h-i). Furthermore, photostimulation of LH terminals from NAc D₁R neurons induced wakefulness. Taken together, these data indicated that NAc^{D₁R}→LH circuit was involved in arousal control partially through disinhibiting LH wake-promoting orexin neurons.

Line 318: The authors cite a publication in press showing that NAc D₂R-expressing neurons strongly promote sleep. They should develop the results obtained in this study. This is indeed important in order to get the whole picture of the role of the NAc neurons in sleep.

Response:

As requested, we carried out a complementary study in D₂R neurons by using D₂R-Cre mice and chemogenetic methods to activate or inhibit NAc D₂R-expressing neurons. The results showed that chemogenetic inhibition of NAc D₂R neurons induced wakefulness which is similar to the result of NAc D₁R neuron activation, while activation of NAc D₂R neurons increased NREM sleep.

Based on these findings, we created a new figure (Figure 8, Pages 44-45, lines 1046-1072) and added texts to the revised manuscript (Page 12, lines 321-339). Our new results are in line with a recent paper that NAc A_{2A}R/D₂R neurons promote slow wave sleep by using adenosine A_{2A}R-Cre mice (Oishi et al., 2017), because adenosine A_{2A}Rs are co-expressed with dopamine D₂Rs in the same neurons in the NAc (Lazarus et al., 2011).

In addition, we added a drawing to clarify our hypothesis (Please see supplementary figure 8, pages 54-55, lines 1186-1201)

330-331: rephrasing of the sentence is necessary

Response:

We modified the sentence to “Collectively, inhibitory input from NAc D₁R neurons to the midbrain disinhibits dopamine neurons and promotes wakefulness” in page 13, lines 364-365 of the revised manuscript.

334: on the firing of DA VTA neurons, the article of Dahan et al. (2007) is missing.

Response:

As suggested, we added this paper in the revised manuscript.

340-342: repetitions again in the sentence

Response:

We modified the sentence to “Our data support a model wherein activating an inhibitory projection from the NAc to the midbrain promotes arousal through inhibition of GABA neurons, which causes disinhibition of DA neurons to increase DA release in the NAc. This might be a positive feedback process, because the released DA acts on D₁R and D₂R to excite D₁R neurons and inhibit D₂R neurons”. Please see page 14, lines 374-378 in the revised manuscript.

Line 343-344: the information on the neuronal target of adenosine in the NAc as published by Lazarus et al. is missing.

Response:

As suggested, we added this paper in the revised manuscript.

345-361: Discussion on the LH is focusing on GABAergic neurons involved in waking. However, this is relatively a new concept whereas there are many publications showing the presence of GABAergic neurons involved in sleep including a very recent one in Nature about ZI neurons. These publications need to be discussed. Missing are also publications demonstrating that a projection from the NAc to LH neurons exists. This is important since many of the fibers there are just passing in the

MFB.

Response:

Thank you for the suggestion. We discussed more about a subpopulation ZI GABAergic (lhx6+) neurons in sleep-regulation in revised manuscript (pages 14-15, lines 398-407) as below.

Discussion: Recently, Liu et al. (2017) identified a GABAergic subpopulation of neurons (Lhx6+) in the ventral zona incerta (ZI), which is adjacent to the LH, promote sleep. They also conducted a retrograde tracing study and confirmed a monosynaptic projection from the NAc to ZI Lhx6+ neurons. We examined IPSCs in ZI neurons following optogenetic stimulation of NAc D₁R neurons terminals in brain slices. Indeed, we observed sparse ChR2 positive fibers around lhx6+ neurons in the ZI (the image below), but we did not detect functional connection (n = 7 cells from 2 mice, data not shown), suggesting that NAc^{D₁R}→ZI pathway might not be crucial for arousal regulation of NAc D₁R neurons.

Several publications demonstrate that anatomical and functional projections from the NAc to LH neurons exist.

- 1) Anatomical observations using anterograde and retrograde tracing methods have shown that NAc projection neurons send axons to the LH (Heimer et al., 1991; Zahm and Heimer, 1993).**
- 2) Pharmacological findings support the proposed NAc to LH circuit: increased feeding following NAcSh inhibition is prevented by**

concomitant infusion of a GABA_A R agonist into the LH (Maldonado-Irizarry et al., 1995; Urstadt et al., 2013).

- 3) *In vivo* single-unit recording revealed that photostimulation of NAcSh fibers modulated neuronal activity in the LH (Prado et al., 2016).**
- 4) Using optogenetic assisted mapping approach, scientists found that NAc projection neurons inhibited LH neurons (Larson et al., 2015; O'Connor et al., 2015).**

Together, these data suggest that NAc neurons really send projections to the LH neurons. We cited related publications in the text of revised manuscript (Line 271).

Line 362 They state that the NAc-VP pathway is also important for sleep without developing the arguments. Again, this is important to figure out which projection does what.

Response:

We thank the important comments. Although both NAc D₁R and D₂R-expressing neurons innervate the VP, quantitative analysis revealed that less than 3% colocalization of D₁- and D₂-expressing fibers in the VP, strongly indicating separate D₁- and D₂-neurons projections to VP (Kupchik et al., 2015). In our study, we think that activation of NAc ^{D₁R}→VP pathway promotes arousal, whereas activation of NAc ^{A₂AR}→VP pathway induced sleep that has been confirmed by Oishi (Oishi et al., 2017). As the discussion mentioned (page 15, lines 408-417), NAc D₁R and D₂R neurons may target different subsets of VP neurons. However, we don't know difference between connections of NAc ^{D₁R}→VP and NAc ^{A₂AR}→VP, and the circuitry by which VP neurons regulate sleep and wakefulness remains largely unresolved. Thus, our next work will focus on different inputs from NAc D₁R and D₂R neurons to the VP. Our preliminary data showed that VP cell types played distinct roles in sleep-wake regulation.

In addition, we added a drawing to show our hypothesis (Please see supplementary figure 8, pages 54-55, lines 1186-1201) in the revised manuscript.

Overall, the discussion needs to provide a clearer picture on the role of the different NAc neurons and projections. They should also discuss the role of interconnections between the two populations as reported in their supplementary Fig. 5. A drawing would be particularly helpful in that matter.

Response:

As suggested, we added a drawing to clarify our hypothesis (Please see supplementary figure 8, pages 54-55, lines 1186-1201) in the revised manuscript.

Supplementary Figure 8. Putative neural circuitry for NAc D₁R neurons controlling wakefulness. Within the NAc, there exists collateral inhibition between D₁R neurons and D₂R neurons. NAc D₁R neurons may directly inhibit sleep-promoting D₂R neurons via local circuits. Both D₁R- and D₂R -neurons in the NAc project to the VP, and NAc D₂R to VP pathway has been considered to be involved in sleep control. The role of NAc D₁R to VP pathway in sleep-wake regulation is needed to uncover. Activation of NAc D₁R neurons may excite LH orexin neurons and promotes wakefulness through disinhibition of orexin neurons. Inhibitory input from NAc D₁R neurons mainly target GABA neurons in the ventral midbrain (VTA/SNc), which may lead to disinhibition of wake-promoting DA neurons. VTA DA neurons also send dopamine projections to the NAc, thus the NAc and VTA are reciprocally connected, forming a

NAc/GABA-midbrain/GABA-VTA/DA-NAc/GABA loop that controls wakefulness.
DA: dopamine; GAD: glutamate decarboxylase; LH: lateral hypothalamus; NAc: nucleus accumbens; OX: orexin; SNc: substantial nigra pars compacta; VP: ventral pallidum; VTA: ventral tegmental area.

Reviewer #3 (Remarks to the Author):

The study is intensive and extensive using various cutting edge techniques, and the results appeared to be reliable. However, it was difficult to understand the functional importance of the work.

I understand from the data presented that NAc D1R neuron circuits are essential for the induction and maintenance of wakefulness, but the manuscript does not address anything about how NAc D1R neuron circuits are physiologically and pathophysiology important. The abstract is merely a list of the findings and does not address functional importance of the NAc D1R neuron circuits. Most of the experiments, except in vivo fiber photometry and electron microscopy experiments, are likely to be supraphysiological and do not provide much information regarding how these NAc D1R neuron circuits coordinate with other systems to regulate sleep and wakefulness.

Response:

We thank the comments. Our fiber photometry data showed arousal-dependent increases in population activity of NAc D₁R neurons in freely moving mice. Next, we used optogenetic and chemogenetic methods to modulate the neuronal activity of NAc D₁R neurons and found that activation of NAc D₁R neurons initiated and prolonged arousal, with decreased food intake and normal locomotor activity. Although optogenetic and chemogenetic activation may be supraphysiological, we introduced chemogenetic inhibition and found that inhibition of NAc D₁R neurons increased sleep, with more nest building behaviors. Nest building is a behavior that mice purposefully engage in when they intend to fall asleep, suggesting that NAc D₁R neuron activity need to be suppressed for animals to prepare for sleep. Therefore, NAc D₁R neurons are essential for arousal controls

By using patch-clamp, viral tracing, immunohistochemistry, and electron microscopy, we revealed two important pathways of NAc D₁R neurons controlling wakefulness: the midbrain and the LH pathways. NAc D₁R neurons

preferentially targeted GABA neurons in the midbrain, activating NAc D_{1R} →midbrain pathway promotes arousal probably through disinhibition of wake-promoting DA neurons in the VTA/SNc. Then, we provided evidence that NAc D_{1R} →LH circuit was involved in arousal control partially through disinhibition of LH wake-promoting orexin neurons.

Combined with the previous study by De Lecea (Eban-Rothschild et al., 2016), our results further demonstrated that the NAc and VTA are reciprocally connected, forming a NAc/GABA-midbrain/GABA-VTA/DA-NAc/GABA loop that controls wakefulness.

Both D1R and D2R neuron exist in the NAc, and D1R and D2R-expressing neurons play complementary and sometimes opposing roles in higher brain functions. A recent research reported that direct stimulation of the NAc A2AR/D2R-expressing neurons induced remarkable non-rapid eye movement (non-REM, NREM) sleep.

These taken together with the fact that the endogenous molecule mediating the NAc D2R and D1R mediating effects is DA, it is important to study how these two DA receptive systems are regulated to control physiological sleep/wake and study how disruptions of these systems affect sleep and other behavioral states.

Response:

To clearly identify the role of two NAc neuronal subtypes in sleep-wake regulation, we carried out a complementary study in D₂R neurons by using D₂R-Cre mice and chemogenetic methods to activate or inhibit NAc D₂R-expressing neurons. The results showed that chemogenetic inhibition of NAc D₂R neurons induced wakefulness, which is similar to the result of NAc D₁R neuron activation, while activation of NAc D₂R neurons increased NREM sleep. Taken together, the results indicated NAc D₁R- and D₂R- neurons exerted an opposite role in regulating sleep-wake behaviors.

Based on these findings, we created Figure 8 (Pages 44-45, lines 1046-1072) and added texts to the revised manuscript (Page 12, lines 321-339). Our newly

added results are in line with a recent paper that NAc A_{2A}R/D₂R neurons promote slow wave sleep by using adenosine A_{2A}R-Cre mice (Oishi et al., 2017), because adenosine A_{2A}Rs are co-expressed with dopamine D₂Rs in the same neurons in the NAc (Lazarus et al., 2011).

In addition to the DA afferents, NAc neurons receive glutamatergic inputs from cortex and subcortex regions, as well as inhibitory inputs from local interneurons and collateral inputs from other medium spiny neurons (Floresco, 2015; Lazarus et al., 2012). Thus, the endogenous molecule mediating the effects of NAc D₂R and D₁R neurons is not only just DA, but also glutamate and GABA. Besides, adenosine is an obvious candidate for activating NAc D₂R positive neurons because excitatory adenosine A_{2A}R were colocalized with D₂Rs.

Therefore, the NAc has the capability to integrate locomotion with motivational behavior through dopaminergic inputs, contextual content from the hippocampus, and emotional information from the amygdala.

In our study, we found that NAc D₁R- and D₂R neurons play the opposite role in sleep-wake regulation, while other distinct roles in reward, sensitization and feeding behaviors are also reported by previous studies (Bock et al., 2013; Lobo et al., 2010; O'Connor et al., 2015; Smith et al., 2013). A balance between these two subpopulations in the NAc is likely necessary for sleep-wake behavior and other behavioral states.

The authors concluded that NAc D₁R-expressing neurons are essential in controlling wakefulness and are involved in physiological arousal via the LH and midbrain circuits, suggesting that the NAc should be considered as a potential target area for therapy in neuropsychiatric disorders with sleep-wake alterations.

This conclusion does not hold much in terms of functional significance and no data presented suggested that the NAc should be considered as a potential target area for therapy in neuropsychiatric disorders with sleep-wake alterations.

Response:

Many studies have shown that NAc dysfunction causes numerous neurological disorders including depression, anxiety disorders, addictive behaviors and compulsive disorders (Lobo et al., 2012; Lobo and Nestler, 2011; Salgado and Kaplitt, 2015; Vetrivelan et al., 2010; Zarrindast and Khakpai, 2015). **These neurological disorders have been reported to associate with sleep-wake alterations** (Abbott and Videnovic, 2016; O'Sullivan et al., 2008; Shulman et al., 2001). **Our study demonstrated that the NAc is important in regulating sleep and wakefulness, and inhibitory NAc projections to the VTA/SNc and the LH, which are shown to be involved in neuropsychiatric behaviors, such as motivation, reward and compulsive feeding** (Bocklisch et al., 2013; Larson et al., 2015; O'Connor et al., 2015).

Our results uncover the distinct roles of NAc D₁R and D₂R neurons in sleep-wake regulation, and NAc D₁R neurons modulate physiological and functional wakefulness mainly through the VTA/SNc and the LH pathways.

Considering the importance of the NAc- VTA/SNc and -LH pathways in physiological and pathophysiological functions, we thought that the NAc may be considered as a potential target area for therapy in neuropsychiatric disorders with sleep-wake alterations.

References:

- Abbott, S.M., and Videnovic, A. (2016). Chronic sleep disturbance and neural injury: Links to neurodegenerative disease. *Nat Sci Sleep* 8, 55-61.
- Bock, R., Shin, J.H., Kaplan, A.R., Dobi, A., Markey, E., Kramer, P.F., Gremel, C.M., Christensen, C.H., Adrover, M.F., and Alvarez, V.A. (2013). Strengthening the accumbal indirect pathway promotes resilience to compulsive cocaine use. *Nat. Neurosci.* 16, 632-638.
- Bocklisch, C., Pascoli, V., Wong, J.C., House, D.R., Yvon, C., de Roo, M., Tan, K.R., and Luscher, C. (2013). Cocaine disinhibits dopamine neurons by potentiation of GABA transmission in the ventral tegmental area. *Science* 341, 1521-1525.
- Eban-Rothschild, A., Rothschild, G., Giardino, W.J., Jones, J.R., and de Lecea, L. (2016). VTA dopaminergic neurons regulate ethologically relevant sleep-wake behaviors. *Nat. Neurosci.* 19, 1356-1366.
- Floresco, S.B. (2015). The nucleus accumbens: An interface between cognition, emotion, and action. *Annu Rev Psychol* 66, 25-52.
- Heimer, L., Zahm, D.S., Churchill, L., Kalivas, P.W., and Wohltmann, C. (1991). Specificity in the

- projection patterns of accumbal core and shell in the rat. *Neuroscience* 41, 89-125.
- Herrera, C.G., Cadavieco, M.C., Jego, S., Ponomarenko, A., Korotkova, T., and Adamantidis, A. (2015). Hypothalamic feedforward inhibition of thalamocortical network controls arousal and consciousness. *Nat. Neurosci.* 19, 290-298.
- Kupchik, Y.M., Brown, R.M., Heinsbroek, J.A., Lobo, M.K., Schwartz, D.J., and Kalivas, P.W. (2015). Coding the direct/indirect pathways by D1 and D2 receptors is not valid for accumbens projections. *Nat. Neurosci.* 18, 1230-1232.
- Larson, E.B., Wissman, A.M., Loriaux, A.L., Kourrich, S., and Self, D.W. (2015). Optogenetic stimulation of accumbens shell or shell projections to lateral hypothalamus produce differential effects on the motivation for cocaine. *J. Neurosci.* 35, 3537-3543.
- Lazarus, M., Huang, Z.L., Lu, J., Urade, Y., and Chen, J.F. (2012). How do the basal ganglia regulate sleep-wake behavior? *Trends Neurosci.* 35, 723-732.
- Lazarus, M., Shen, H.Y., Cherasse, Y., Qu, W.M., Huang, Z.L., Bass, C.E., Winsky-Sommerer, R., Semba, K., Fredholm, B.B., and Boison, D., *et al.* (2011). Arousal effect of caffeine depends on adenosine A2A receptors in the shell of the nucleus accumbens. *J. Neurosci.* 31, 10067-10075.
- Liu, K., Kim, J., Kim, D.W., Stephanie, Z.Y., Bao, H., Denaxa, M., Lim, S.A., Kim, E., Liu, C., and Wickersham, I.R., *et al.* (2017). Corrigendum: Lhx6-positive GABA-releasing neurons of the zona incerta promote sleep. *Nature* 550, 548.
- Lobo, M.K., Covington, H.R., Chaudhury, D., Friedman, A.K., Sun, H., Damez-Werno, D., Dietz, D.M., Zaman, S., Koo, J.W., and Kennedy, P.J., *et al.* (2010). Cell type-specific loss of BDNF signaling mimics optogenetic control of cocaine reward. *Science* 330, 385-390.
- Lobo, M.K., Nestler, E.J., and Covington, H.R. (2012). Potential utility of optogenetics in the study of depression. *Biol Psychiatry* 71, 1068-1074.
- Lobo, M.K., and Nestler, E.J. (2011). The striatal balancing act in drug addiction: Distinct roles of direct and indirect pathway medium spiny neurons. *Front Neuroanat* 5, 41.
- Maldonado-Irizarry, C.S., Swanson, C.J., and Kelley, A.E. (1995). Glutamate receptors in the nucleus accumbens shell control feeding behavior via the lateral hypothalamus. *J. Neurosci.* 15, 6779-6788.
- O'Connor, E.C., Kremer, Y., Lefort, S., Harada, M., Pascoli, V., Rohner, C., and Luscher, C. (2015). Accumbal D1R neurons projecting to lateral hypothalamus authorize feeding. *Neuron* 88, 553-564.
- Oishi, Y., Xu, Q., Wang, L., Zhang, B.J., Takahashi, K., Takata, Y., Luo, Y.J., Cherasse, Y., Schiffmann, S.N., and de Kerchove, D.A., *et al.* (2017). Slow-wave sleep is controlled by a subset of nucleus accumbens core neurons in mice. *Nat Commun* 8, 734.
- O'Sullivan, S.S., Williams, D.R., Gallagher, D.A., Massey, L.A., Silveira-Moriyama, L., and Lees, A.J. (2008). Nonmotor symptoms as presenting complaints in Parkinson's disease: A clinicopathological study. *Mov Disord* 23, 101-106.
- Prado, L., Luis-Islas, J., Sandoval, O.I., Puron, L., Gil, M.M., Luna, A., Arias-Garcia, M.A., Galarraga, E., Simon, S.A., and Gutierrez, R. (2016). Activation of glutamatergic fibers in the anterior NAc shell modulates reward activity in the aNAcSh, the lateral hypothalamus, and medial prefrontal cortex and transiently stops feeding. *J. Neurosci.* 36, 12511-12529.
- Salgado, S., and Kaplitt, M.G. (2015). The nucleus accumbens: A comprehensive review. *Stereotact Funct Neurosurg* 93, 75-93.
- Sano, H., and Yokoi, M. (2007). Striatal medium spiny neurons terminate in a distinct region in the lateral hypothalamic area and do not directly innervate orexin/hypocretin- or melanin-concentrating hormone-containing neurons. *J. Neurosci.* 27, 6948-6955.

- Shulman, L.M., Taback, R.L., Bean, J., and Weiner, W.J. (2001). Comorbidity of the nonmotor symptoms of Parkinson's disease. *Mov Disord* *16*, 507-510.
- Smith, R.J., Lobo, M.K., Spencer, S., and Kalivas, P.W. (2013). Cocaine-induced adaptations in D1 and D2 accumbens projection neurons (a dichotomy not necessarily synonymous with direct and indirect pathways). *Curr. Opin. Neurobiol.* *23*, 546-552.
- Urstadt, K.R., Kally, P., Zaidi, S.F., and Stanley, B.G. (2013). Ipsilateral feeding-specific circuits between the nucleus accumbens shell and the lateral hypothalamus: Regulation by glutamate and GABA receptor subtypes. *Neuropharmacology* *67*, 176-182.
- Venner, A., Anaclet, C., Broadhurst, R.Y., Saper, C.B., and Fuller, P.M. (2016). A novel population of Wake-Promoting GABAergic neurons in the ventral lateral hypothalamus. *Curr. Biol.* *26*, 2137-2143.
- Vetrivelan, R., Qiu, M.H., Chang, C., and Lu, J. (2010). Role of Basal Ganglia in sleep-wake regulation: Neural circuitry and clinical significance. *Front Neuroanat* *4*, 145.
- Yuan, X.S., Wang, L., Dong, H., Qu, W.M., Yang, S.R., Cherasse, Y., Lazarus, M., Schiffmann, S.N., D'Exaerde, A.K., Li, R.X., and Huang, Z.L. (2017). Striatal adenosine A2A receptor neurons control active-period sleep via parvalbumin neurons in external globus pallidus. *Elife* *6*.
- Zahm, D.S., and Heimer, L. (1993). Specificity in the efferent projections of the nucleus accumbens in the rat: Comparison of the rostral pole projection patterns with those of the core and shell. *J. Comp. Neurol.* *327*, 220-232.
- Zarrindast, M.R., and Khakpai, F. (2015). The modulatory role of dopamine in anxiety-like behavior. *Arch Iran Med* *18*, 591-603.

REVIEWERS' COMMENTS:

Reviewer #1 (Remarks to the Author):

The manuscript has been adequately revised. I have nothing further.

Reviewer #2 (Remarks to the Author):

The authors well answered to my requests.

I have only one comment concerning the drawing of the pathways involved.

I believe this is a very nice addition to the paper.

I believe that a few information should be added to it.

It should be added that adenosine excite the D2-expressing neurons in the NAc.

I should be also shown that DA should excite D2-expressing neurons and inhibit D1-expressing neurons based on our knowledge of the effect of these two receptors.

Do you agree with that? In this case, how we can explain that DA is inducing waking if it excite sleep neurons in the NAc? This should be discussed.

Conversely, how come DA inhibit the D1 wake inducing neurons?

You also show only one pathway from the D2 NAc neurons to the VP, there are no additional projections?

It would be nice to also discuss these points in the text or the legend of the figure.

The specific comments of reviewers and editors are addressed with the point-by-point responses as follows:

Reviewer #1 (Remarks to the Author):

The manuscript has been adequately revised. I have nothing further.

Reviewer #2 (Remarks to the Author):

The authors well answered to my requests.

Response: Thank you very much for the comments.

I have only one comment concerning the drawing of the pathways involved.

I believe this is a very nice addition to the paper.

I believe that a few information should be added to it.

- It should be added that adenosine excites the D2-expressing neurons in the NAc.

Response: As requested, we added this information to the supplementary figure 8 and revised the legend accordingly.

Supplementary Figure 8. Putative neural circuitry for NAc D₁R neurons controlling wakefulness. (a) Within the NAc, there exists collateral inhibition between D₁R neurons and D₂R neurons. NAc D₁R neurons may directly inhibit sleep-promoting D₂R neurons via local circuits. Both D₁R- and D₂R -neurons in the

NAc project to the VP (Kupchik et al., 2015). NAc D₂R/A_{2A}R to VP pathway has been considered to be involved in sleep control (Oishi et al., 2017), whereas NAc D₂R/A_{2A}R to LH and VTA pathways are not important for sleep induction of NAc D₂R/A_{2A}R neurons (Oishi et al., 2017). However, the role of NAc D₁R to VP pathway in sleep-wake regulation is needed to uncover. Activation of NAc D₁R neurons may excite LH orexin neurons and promotes wakefulness through disinhibition of orexin neurons. Inhibitory inputs from NAc D₁R neurons mainly target GABA neurons in the ventral midbrain (VTA/SNc), which may lead to disinhibition of wake-promoting DA neurons. VTA DA neurons also send dopamine projections to the NAc (Eban-Rothschild et al., 2016), thus the NAc and VTA are reciprocally connected, forming a NAc/GABA-midbrain/GABA-VTA/DA-NAc/GABA loop that controls wakefulness. **(b)** NAc D₁R neurons also express adenosine A₁ receptors, whereas D₂R neurons coexpress adenosine A_{2A} receptors. D₁Rs and A_{2A}Rs are primarily coupled to Gs/olf proteins and stimulate the activity of adenylate cyclase, followed by PKA activation via cAMP accumulation. By contrast, D₂Rs and A₁Rs are associated with Gi/o proteins to inhibit cAMP production (Lazarus et al., 2013; Lazarus et al., 2012; Nagai et al., 2016). *In vitro and in vivo* electrophysiological studies indicate that D₁R activation modulates the intrinsic excitability of NAc neurons, and usually increases neuron activities, whereas D₂R activation often attenuates spike activities (Hopf et al., 2003; Perez et al., 2006; West and Grace, 2002). Thus, DA would differentially modulate D₁R- and D₂R- neuron activities through its action on excitatory D₁Rs or inhibitory D₂Rs, both of which are involved in wakefulness. On the other hand, adenosine promotes sleep through the activation of inhibitory A₁Rs and excitatory A_{2A}Rs. A₁R: adenosine A₁ receptor; A_{2A}R: adenosine A_{2A} receptor; D₁R: dopamine D₁ receptor; D₂R: dopamine D₂ receptor; DA: dopamine; GAD: glutamate decarboxylase; LH: lateral hypothalamus; NAc: nucleus accumbens; OX: orexin; SNc: substantial nigra pars compacta; VP: ventral pallidum; VTA: ventral tegmental area.

- It should be also shown that DA should excite D₂-expressing neurons and inhibit D₁-expressing neurons based on our knowledge of the effect of these two receptors.

Do you agree with that? In this case, how we can explain that DA is inducing waking if it excite sleep neurons in the NAc? This should be discussed.

Conversely, how come DA inhibit the D1 wake inducing neurons?

Response: The D₁R is coupled to adenylyate cyclase through G_{s/olf} and activates protein kinase A (PKA), whereas the D₂R inhibits adenylyate cyclase through G_{i/o} (Herve et al., 1993; Nagai et al., 2016). *In vitro and in vivo* electrophysiological studies indicate that D₁R activation modulates the intrinsic excitability of NAc neurons, and usually increases neuron activities, whereas D₂R activation often attenuates spike activities (Hopf et al., 2003; Perez et al., 2006; West and Grace, 2002). Thus, DA would differentially modulate neuronal activity of NAc two subpopulations through its action on excitatory D₁Rs or inhibitory D₂Rs, both of which are involved in wakefulness.

We added above discussion into the legend of supplementary figure 8.

- You also show only one pathway from the D2 NAc neurons to the VP, there are no additional projections?

It would be nice to also discuss these points in the text or the legend of the figure.

Response: Anatomical studies show that NAc D₂R/A_{2A}R neurons mainly project to the VP, and send sparse fibers to the LH and VTA (Francis et al., 2015; O'Connor et al., 2015; Zhang et al., 2013). However, optogenetic-assisted circuit mapping studies find that light-evoked IPSCs in VP cells, but not in VTA neurons (Kupchik et al., 2015; Oishi et al., 2017). A weak functional connection between NAc D₂R/A_{2A}R neurons and LH cells is detected *in vitro* (O'Connor et al., 2015; Oishi et al., 2017). *In vivo* optical stimulation of NAc A_{2A}R neurons terminals in the VP, but not the LH and the VTA, produces NREM sleep. Thus, only NAc D₂R/A_{2A}R to VP pathway is important in sleep induction.

As requested, we added these major points to the legend of supplementary figure 8.

References:

- Eban-Rothschild, A., Rothschild, G., Giardino, W.J., Jones, J.R., and de Lecea, L. (2016). VTA dopaminergic neurons regulate ethologically relevant sleep-wake behaviors. *Nat. Neurosci.* *19*, 1356-1366.
- Francis, T.C., Chandra, R., Friend, D.M., Finkel, E., Dayrit, G., Miranda, J., Brooks, J.M., Iniguez, S.D., O'Donnell, P., Kravitz, A., and Lobo, M.K. (2015). Nucleus accumbens medium spiny neuron subtypes mediate depression-related outcomes to social defeat stress. *Biol. Psychiatry.* *77*, 212-222.
- Herve, D., Levi-Strauss, M., Marey-Semper, I., Verney, C., Tassin, J.P., Glowinski, J., and Girault, J.A. (1993). G(olf) and Gs in rat basal ganglia: Possible involvement of G(olf) in the coupling of dopamine D1 receptor with adenylyl cyclase. *J. Neurosci.* *13*, 2237-2248.
- Hopf, F.W., Cascini, M.G., Gordon, A.S., Diamond, I., and Bonci, A. (2003). Cooperative activation of dopamine D1 and D2 receptors increases spike firing of nucleus accumbens neurons via G-protein betagamma subunits. *J. Neurosci.* *23*, 5079-5087.
- Kupchik, Y.M., Brown, R.M., Heinsbroek, J.A., Lobo, M.K., Schwartz, D.J., and Kalivas, P.W. (2015). Coding the direct/indirect pathways by D1 and D2 receptors is not valid for accumbens projections. *Nat. Neurosci.* *18*, 1230-1232.
- Lazarus, M., Chen, J., Urade, Y., and Huang, Z. (2013). Role of the basal ganglia in the control of sleep and wakefulness. *Curr. Opin. Neurobiol.* *23*, 780-785.
- Lazarus, M., Huang, Z.L., Lu, J., Urade, Y., and Chen, J.F. (2012). How do the basal ganglia regulate sleep-wake behavior? *Trends Neurosci.* *35*, 723-732.
- Nagai, T., Yoshimoto, J., Kannon, T., Kuroda, K., and Kaibuchi, K. (2016). Phosphorylation signals in striatal medium spiny neurons. *Trends Pharmacol. Sci.* *37*, 858-871.
- O'Connor, E.C., Kremer, Y., Lefort, S., Harada, M., Pascoli, V., Rohner, C., and Luscher, C. (2015). Accumbal D1R neurons projecting to lateral hypothalamus authorize feeding. *Neuron* *88*, 553-564.
- Oishi, Y., Xu, Q., Wang, L., Zhang, B.J., Takahashi, K., Takata, Y., Luo, Y.J., Cherasse, Y., Schiffmann, S.N., and de Kerchove, D.A., *et al.* (2017). Slow-wave sleep is controlled by a subset of nucleus accumbens core neurons in mice. *Nat. Commun.* *8*, 734.
- Perez, M.F., White, F.J., and Hu, X.T. (2006). Dopamine D(2) receptor modulation of K(+) channel activity regulates excitability of nucleus accumbens neurons at different membrane potentials. *J. Neurophysiol.* *96*, 2217-2228.
- West, A.R., and Grace, A.A. (2002). Opposite influences of endogenous dopamine D1 and D2 receptor activation on activity states and electrophysiological properties of striatal neurons: Studies combining in vivo intracellular recordings and reverse microdialysis. *J. Neurosci.* *22*, 294-304.
- Zhang, J.P., Xu, Q., Yuan, X.S., Cherasse, Y., Schiffmann, S.N., de Kerchove, D.A., Qu, W.M., Urade, Y., Lazarus, M., Huang, Z.L., and Li, R.X. (2013). Projections of nucleus accumbens adenosine A2A receptor neurons in the mouse brain and their implications in mediating sleep-wake regulation. *Front. Neuroanat.* *7*, 43.